# Observer-Side Diagnosis of Prompt-Induced Behavioral Shifts in Large Language Models:
# A Diagnostic Vocabulary and a Pilot Japanese/English Politeness Probe

## Abstract

Small prompt fragments can change not only an intended surface property of a response, such as register, but also observable markers of uncertainty, conditionality, and scope. This paper introduces an *observer-side* diagnostic vocabulary for reporting such prompt-induced behavioral shifts under black-box access. The vocabulary organizes prompt effects into four macro-groups—Framing, Reasoning, Expression, and Epistemic control—and instantiates them as the Z-model, an auditable 11-axis reference basis for describing prompt–response interactions. The reference basis is a reporting device rather than a latent-state estimator, causal model, or predictive control method. Empirically, we deliberately narrow the evidence-bearing claim to a pilot Japanese/English politeness probe: one Expression-oriented cue ("politely" / 「丁寧に」 ) is tested for secondary shifts in surface markers associated with epistemic stance and scope. Under a matched protocol with five benign technical topics and 250 samples per language-condition, we report a revision-specified four-endpoint primary family, use topic-aware hierarchical bootstrap confidence intervals, report Benjamini–Hochberg correction across the primary tests, and add raw-count checks to guard against length-normalization artifacts. The robust automatic findings are localized within the API-served pilot: English polite prompts increase epistemic hedging, while Japanese polite prompts increase conditional framing; both remain positive in raw counts. A pinned open-weight model-sensitivity comparison shows the English hedging direction more clearly than the Japanese conditional-framing direction, so the Japanese result is treated as model- and snapshot-dependent rather than cross-model validation. Other shifts are reported as directional or exploratory. A supplemental audience-framing probe instantiates a second Framing→Reasoning path, now recomputed from raw audience-framing generations; novice/non-specialist framing increases analogy-oriented and explanatory-scaffolding markers in both languages, while conditional markers do not uniformly increase. A language-qualified direct-reader check and a separate LLM-judge rubric check are reported as exploratory interpretive context rather than confirmatory semantic validation. Together with the measurement-theory framing, they make visible where automatic marker counts align or fail to align with reader- or rubric-based judgments; they are not used as evidence for or against semantic constructs. The automatic findings should therefore be read as marker-level audit signals, not as evidence of reader-perceived semantic change or validation of the 11-axis reference basis. Overall, the contribution is a scoped diagnostic vocabulary plus a reproducible pilot protocol for auditing selected cross-group prompt effects; it does not validate the full 11-axis framework or recover hidden model states.

## 1 Introduction

Small prompt edits can change more than wording. A fragment that appears to target register or politeness can also shift how a model marks uncertainty, states conditions, lists alternatives, or presents closure. Such

effects matter for interaction-level reliability, but they are not well captured by task-level prompt evaluation alone, where the main target is usually accuracy, benchmark performance, or output quality (Liu et al., 2023; White et al., 2023; Sclar et al., 2024; Razavi et al., 2025; Zhuo et al., 2024). This paper studies these shifts as *observable behavioral redistribution* under matched black-box perturbations.

**Scope of the empirical claim.** The empirical component of this paper is a *pilot diagnostic case study*, not a validation of a full theory of prompting or a semantic-perception model. We focus on one high-leverage pathway: an Expression-oriented politeness cue ("politely" /「丁寧に」) and its secondary effects on epistemic- and scope-related surface markers in Japanese and English. The probe is intentionally narrow: five benign technical topics, one API-served execution window for the main politeness probe, one pinned open-weight comparison path, and lightweight lexical/structural indicators. All empirical claims are therefore stated as within-language, within-protocol distributional deltas in observable output features. We do not claim improved factuality, safer behavior, calibrated uncertainty, or cross-lingual typological generality.

**Observer-side vocabulary.** To describe such effects, we introduce a macro-group diagnostic vocabulary for prompt-induced behavior: **Framing**, **Reasoning**, **Expression**, and **Epistemic control**. The Z-model instantiates this vocabulary as an auditable 11-axis reference basis for reporting and comparison under black-box access. Where the term *latent* appears, it is used only in an observer-side analytical sense: it summarizes unobserved degrees of freedom inferred from prompt–response behavior, without assuming identifiable model-internal variables. For reference, the compact observer-side block is

$$p \in \mathcal{P}, \qquad \boldsymbol{z}(p) = f_{\mathrm{enc}}(p) \in \mathbb{R}^{11}, \qquad y \sim q(\cdot \mid \boldsymbol{z}(p)), \qquad r(y) \in \mathbb{R}^m, \tag{1}$$

where $r(y)$ is the observable indicator vector used for diagnosis. Equation (1) is bookkeeping notation; it is not an estimator, causal graph, or claim about internal representations.

**Motivating example with gloss.** Consider two Japanese prompts that differ by an added agreement-seeking fragment:

> (A)「慎重に検討してください。」 *Shinchou ni kentou shite kudasai.* Literal gloss: "Please examine carefully."
>
> (B)「慎重に検討して説明してください。結論は正しいですよね？」 *Shinchou ni kentou shite setsumei shite kudasai. Ketsuron wa tadashii desu yo ne?* Literal gloss: "Please examine carefully and explain. The conclusion is correct, right?"

The point is not that (B) is a natural or recommended prompt. It is that a small surface addition can alter observable uncertainty, conditionality, and closure markers even when the underlying topic is unchanged. The Z-model provides terminology for reporting such primary and secondary shifts.

**Contributions.** The paper makes three scoped contributions. First, it proposes an observer-side vocabulary for reporting prompt-induced secondary shifts across macro-groups, while explicitly rejecting latent identifiability or model-internal interpretation. Second, it provides a pilot Japanese/English politeness-probe protocol with a revision-specified primary endpoint family, topic-aware hierarchical bootstrap, multiple-comparison handling, and a split between additive discourse markers and genuine contrastive-alternative markers. Third, it adds two support layers around the pilot rather than two new primary claims: a raw-data-derived supplementary Framing→Reasoning audience-framing probe, summarized in the main text to show that a second macro-group path can be audited, and a language-qualified direct-reader check, together with a completed rubric-based LLM-judge check over the full main-probe sample, that transparently reports where automatic marker quantity does or does not align with reader- or judge-rated uncertainty, scope/sufficiency, and explanatory scaffolding.

**Roadmap.** Section 2 situates the work in prompt sensitivity, instruction-conflict, calibration, and pragmatics research. Section 3 defines the diagnostic vocabulary, reports the pilot politeness probe, summarizes the supplemental Framing→Reasoning probe, and describes the language-qualified reader check. Section 4

discusses the supported interpretation, remaining limitations, and how the vocabulary can be used as an audit scaffold. Appendices contain the expanded examples, regex dictionaries, reproducibility details, open-weight sensitivity comparison, indicator ablations, exploratory probes, and reader-check materials.

## 2 Background and Related Work

This paper connects four adjacent literatures: prompt engineering, instruction-conflict benchmarking, calibration and hallucination, and language-specific prompting. The review below focuses on results that help explain how prompt variation, alignment constraints, and pragmatic form shape observable response behavior. Rather than re-surveying prompt optimization broadly, the goal is to situate the present interaction-level diagnostic perspective within these neighboring lines of work.

### 2.1 Prompt Sensitivity and Behavioral Evaluation

Recent studies have shown that LLM outputs can be highly sensitive to seemingly minor prompt variations, including wording, formatting, role framing, and reasoning instructions. Prompting work such as Chain-of-Thought prompting demonstrates that small structural changes can substantially alter reasoning behavior and task outcomes (Wei et al., 2022; Kojima et al., 2022). More broadly, behavioral evaluation in NLP has emphasized that models should be assessed not only by aggregate end-task accuracy but also through controlled perturbation-based probes. For example, CheckList introduced capability-oriented behavioral testing templates for systematically surfacing model weaknesses under targeted perturbations (Ribeiro et al., 2020).

Taken together, prior studies show that prompt variation can materially affect model behavior, typically assessed through task outcomes, correctness, or output quality (Sclar et al., 2024; Zhuo et al., 2024; Razavi et al., 2025). The present work builds on this literature but shifts the unit of analysis from task-level outcomes to interaction-level behavioral redistribution under controlled A/B perturbations. The coupled cross-dimensional effects of interest are later operationalized as *secondary marker redistribution.*

### 2.2 Instruction Tuning, Alignment, and Instruction Conflicts

Instruction tuning and human-feedback-based alignment fundamentally shape how LLMs respond to prompts. InstructGPT demonstrated that fine-tuning with human demonstrations and reinforcement learning from human feedback (RLHF) yields models that follow instructions more reliably and are preferred by human evaluators (Ouyang et al., 2022). The Flan collection further systematized instruction tuning data and prompt-format mixtures, improving generalization and reducing the need for downstream fine-tuning (Chung et al., 2022).

A practical consequence of alignment is that prompts often encode *multiple constraints* (style, safety, scope, hedging, and format) that can become partially conflicting in realistic workflows. Recent evaluation work has explicitly focused on this regime: He et al. (2025) proposed ConInstruct to benchmark conflict detection and resolution when instructions contain incompatible constraints, highlighting that models may detect conflicts yet fail to surface them to users. This setting is closely related to the observer-side secondary-redistribution view used here: a single prompt fragment (e.g., politeness or safety cues) can unintentionally reweight other behavioral objectives, changing the resulting epistemic presentation, scope, or reasoning depth.

While related, *secondary redistribution* is distinguished from classic instruction-conflict settings along three axes. First, instruction-conflict benchmarks typically define conflict as two or more explicitly stated constraints that may be mutually incompatible; secondary redistribution is defined at the fragment level via systematic off-target marker shifts (Eq. (4)) even in the absence of an explicit conflict. Second, conflict resolution asks whether a model detects, surfaces, or prioritizes constraints, whereas our diagnostic goal is to characterize cross-dimensional *redistribution* in observable indicators under matched A/B perturbations. Third, the Z-model gives a composition-level reporting expectation: fragments can combine non-additively in structured ways, operationalized by nonzero interaction effects ($d_I \neq 0$) in the factorial probe (Section 3.4).

### 2.3 Calibration, Hallucination, and Epistemic Presentation

A growing body of work examines calibration and uncertainty properties of modern LLMs, analyzing how pretraining and alignment stages affect confidence reliability and proposing output-level methods such as post-hoc calibration, uncertainty-aware scoring, or multi-sample aggregation (Geng et al., 2024; Desai & Durrett, 2020; Kadavath et al., 2022). In parallel, hallucination, fluent but factually incorrect or ungrounded output, remains a major barrier to deployment. Surveys have distinguished factuality from faithfulness hallucinations and reviewed detection and mitigation approaches, including retrieval-augmented generation, self-consistency, and verifier models (Ji et al., 2023; Huang et al., 2025; Tonmoy et al., 2024).

The Z-model is complementary in that it does not propose a new calibration or hallucination-mitigation algorithm. Instead, it introduces an observer-side vocabulary, particularly via $z_{\text{epistemic}}$ and $z_{\text{scope}}$, to reason about how prompts configure epistemic stance and coverage *before* a post-hoc method is applied. This distinction matters for interaction-level reliability: even when factual accuracy is unchanged, changes in *epistemic presentation* (e.g., hedging versus assertiveness, responsibility framing, and refusal versus compliance) can substantially alter downstream user trust and risk.

### 2.4 Pragmatics, Language-Specific Prompting, and Japanese as a Diagnostic Setting

Most prompt engineering and alignment research implicitly assumes English as the primary language; however, prompting behavior is strongly language- and pragmatics-dependent. A recent survey of pragmatic evaluation resources highlighted the breadth of pragmatic phenomena relevant to modern LLMs (e.g., implicature, reference, modality, and social meaning) and the challenges of measuring them reliably (Ma et al., 2025). Cross-lingual studies on politeness have further shown that the effect of prompt politeness on LLM performance differs across languages, including English, Chinese, and Japanese (Yin et al., 2024).

For Japanese specifically, Gan & Mori (2023) analyze sensitivity to prompt templates in Japanese text classification and report large performance swings under small template perturbations. Mikami et al. (2025) constructed a Japanese natural language interface (NLI) dataset for comparatives and observed that model performance can be sensitive to prompt format in zero-shot settings and to example properties in few-shot settings, with difficulties on Japanese-specific phenomena.

Collectively, these studies demonstrated that Japanese prompts can be unusually fragile and that politeness, sentence structure, and other pragmatic-linguistic properties strongly modulate model behavior. However, many previous studies have reported empirical deltas without an interaction-level reporting vocabulary for why certain Japanese fragments (e.g., sentence-final epistemic control or politeness markers) can simultaneously affect style, epistemic stance, reasoning depth, and scope. The Z-model directly addresses this gap. Japanese is treated here as one diagnostic setting in which pragmatic and morphosyntactic cues may make selected coupling patterns easier to observe; we do not infer typological generality from the present probe.

**Linguistic grounding (Japanese).** The Japanese-specific mechanisms discussed in this study are not introduced as ad-hoc properties of LLM outputs, but are grounded in well-established descriptions of Japanese grammar and pragmatics. Japanese is discourse- and topic-oriented, with pervasive argument omission and heavy reliance on contextual inference, as well as a rich system of politeness, honorifics, and sentence-final modality that jointly shape social stance and epistemic commitment. These linguistic properties motivate the treatment of Japanese prompt fragments as a stress test for prompt-induced coupling, helping to explain why minimal surface perturbations can systematically cascade into multiple behavioral dimensions.

**Positioning.** Prior prompt-engineering studies provided catalogs of patterns and best practices, often evaluated by task success. The focus of this study is complementary: prompts are characterized as interaction-level interventions that redistribute multiple behavioral degrees of freedom (e.g., epistemic stance and scope), and cross-dimensional side effects measured as secondary marker redistribution via paired A/B probing are emphasized. This study also distinguishes its setting from factuality or calibration benchmarks: it does not claim to measure truthfulness directly; instead, *epistemic presentation* and *responsibility framing* are measured as observable correlates that can influence downstream reliability judgments.

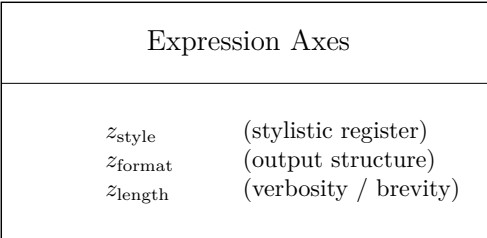
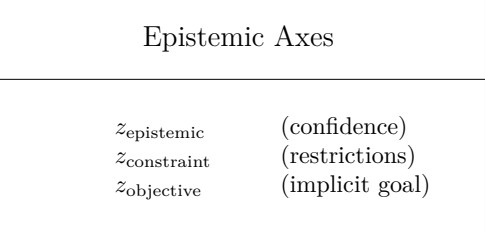

Figure 1: Z-model observer-side reporting axes and their organization into four macro-groups (framing, reasoning, expression, and epistemic control). Each axis represents an observer-side behavioral degree of freedom implicitly configured by a prompt during interaction, rather than a model-internal variable. The finite-dimensional reference basis (instantiated here with 11 axes) is introduced for interpretability, auditability, and operational comparison under black-box access, rather than as a claim of completeness or ontological structure. The axes are not assumed to be independent or orthogonal; apparent overlap reflects empirically recurrent sources of cross-dimensional coupling, which are reported here as secondary marker redistribution. This figure represents an analytical reporting coordinate sheet for reasoning about prompt-induced observable marker shifts, rather than a taxonomy of prompt types or a decomposition of internal model representations.

## 3 Z-model as an Observer-Side Reporting Vocabulary

The Z-model is a *diagnostic vocabulary* for describing prompt-conditioned behavioral tendencies at the interaction level. It is not an identifiable latent-variable model, not a causal model, not an estimator of hidden states, and not a predictive control method. The 11 axes are retained as a pragmatic reference basis because they make primary versus secondary prompt effects easier to report, compare, and audit; the number 11 is not claimed to be minimal, unique, complete, or geometrically meaningful.

Concretely, the vocabulary organizes recurring prompt effects along role, task interpretation, audience assumption, stylistic register, output structure, verbosity, reasoning process, scope of coverage, epistemic stance, constraints, and implicit objectives. These dimensions are introduced to support analysis and reporting, rather than estimation or optimization. They are not assumed to be independent, orthogonal, directly measurable, or model-internal. When directional notation ($\uparrow$ / $\downarrow$) is used for $z_{\text{epistemic}}$, $z_{\text{epistemic}} \uparrow$ denotes higher epistemic commitment (more categorical or definitive statements), whereas $z_{\text{epistemic}} \downarrow$ denotes lower commitment (more explicit uncertainty and hedging).

The key diagnostic idea is *secondary shift*: a prompt fragment with an intended primary target, such as register or format, can produce systematic changes in observable features associated with other dimensions. In this paper, the empirical evidence for such shifts is limited to one pilot Expression→Epistemic/Scope pathway. Figure 1 summarizes the four macro-groups and the 11-axis reference-basis instantiation used as the reporting vocabulary.

### 3.1 Overview of the Z-model and compact observer-side block

An explicit finite-dimensional basis is required not for completeness, but to discuss cross-dimensional coupling, redistribution, and comparison operationally under black-box access. The Z-model is formalized as an

observer-side reporting coordinate sheet for describing behavioral degrees of freedom induced by prompts, intended for analysis rather than statistical estimation.

**Terminological clarification.** The term "coordinate system" is used here in a representational, observer-side sense. It denotes a structured descriptive scheme for organizing directional tendencies in observable output space, without presupposing metric structure, linear embedding, or latent identifiability in a geometric sense. The ambient notation $\mathbb{R}^{11}$ is adopted solely to indicate finite dimensionality and bookkeeping convenience.

Vector notation (e.g., $\Delta z$) expresses directional changes in proxy-aligned dimensions and should not be interpreted as evidence for an underlying continuous latent geometry or internal embedding space. In this sense, the Z-model functions as a structured diagnostic vocabulary instantiated as a reference-basis representation, rather than as a statistical or geometric model of internal states.

Let

$$z = (z_{\text{role}}, z_{\text{task}}, z_{\text{audience}}, z_{\text{style}}, z_{\text{format}}, z_{\text{length}}, z_{\text{process}}, z_{\text{scope}}, z_{\text{epistemic}}, z_{\text{constraint}}, z_{\text{objective}}) \in \mathcal{Z} \subset \mathbb{R}^{11}. \tag{2}$$

where each coordinate represents a distinct behavioral degree of freedom implicitly configured by a prompt. Here, $\mathbb{R}^{11}$ is used purely as a notational convenience to indicate dimensionality, rather than to imply metric structure, continuity, or statistical estimability.

Equation (1) makes explicit the compact observer-side reporting block assumed throughout: prompts configure $z$, $z$ biases the distribution over outputs, and only observable indicators $r(y)$ are measured. All empirical claims in this paper are therefore made at the level of output distributions or proxy expectations under controlled perturbations, not at the level of single-output determinism or identifiable model-internal variables.

The choice of 11 dimensions follows the four macro-groups in Figure 1 (framing, reasoning, expression, and epistemic control), which were sufficient to describe the recurrent degrees of freedom surfaced by the Japanese case studies and the minimal probing protocol. This set is intended as a compact basis for analysis: future work may merge, refine, or extend axes without changing the marker-level definition of secondary redistribution.

Japanese is used in this study as one diagnostic setting: because stance and politeness are often encoded morphosyntactically, selected Expression-level cues may make epistemic- and scope-related marker shifts easier to observe under lightweight black-box probes. The present pilot does not establish typological generality, and analogous analyses in other languages require language-specific indicator design.

### 3.2 Definition of the 11 Observer-Side Reporting Axes

Table 1 is the canonical main-text inventory of the 11 observer-side dimensions. We avoid repeating the same definitions in prose: the table gives the compact definitions used throughout the paper, and extended prompt examples are deferred to the appendix. The dimensions are not assumed to be independent; their overlap is precisely what makes secondary prompt effects auditable under the macro-group vocabulary.

Together, these dimensions form a reference-basis representation through which prompt–model interactions can be organized analytically. The table should be read as a reporting resolution, not as a claim of theoretical uniqueness or completeness.

**Why retain the full 11-axis reference-basis instantiation?** The *conceptual* core of the diagnostic vocabulary comprises the four macro-groups in Figure 1. Nevertheless, this study presents an explicit 11-axis reference-basis instantiation as the *reference basis* for three pragmatic reasons. First, **coverage**: in real prompting, many fragments primarily target Framing or implicit objectives (e.g., role/audience constraints, persuasion vs. critique), and secondary effects often propagate into Reasoning and Epistemic control; without named axes in the reference basis, such patterns are hard to report consistently. Second, **auditability**: a fixed reference basis makes primary vs. secondary shifts explicit and comparable across studies, prompts, and model snapshots under black-box access. Third, **modularity**: the same analysis can be reported at multiple resolutions, including macro-group coarsenings or probe-dependent reductions/rotations (see Figure 2 and Appendix L).

Table 1: 11 observer-side reporting axes of the Z-model (core definitions).

| Axis | Description (concise) |
| --- | --- |
| $z_{\text{role}}$ | Adopted persona or social identity (e.g., expert, tutor, critic), influencing perspective-taking and epistemic authority. |
| $z_{\text{task}}$ | Internal task schema inferred by the model (e.g., summarization, explanation, critique, translation). |
| $z_{\text{audience}}$ | Assumed target reader and expertise level, modulating terminology choice and explanatory depth. |
| $z_{\text{style}}$ | Stylistic register including politeness, formality, and affect; pragmatically dense in Japanese. |
| $z_{\text{format}}$ | Structural organization of the output, such as bullet points, tables, JSON, or stepwise formats. |
| $z_{\text{length}}$ | Expected verbosity or brevity of the response, often constraining other observer-side axes. |
| $z_{\text{process}}$ | Reasoning structure, including step-by-step, contrastive, or outline-first reasoning. |
| $z_{\text{scope}}$ | Breadth of conceptual coverage; in empirical use, enumerative breadth is separated from reader-perceived sufficiency. |
| $z_{\text{epistemic}}$ | Epistemic stance expressed by the model, including confidence, hedging, and acknowledgment of uncertainty. |
| $z_{\text{constraint}}$ | Explicit restrictions or prohibitions imposed by the prompt, such as forbidden content or stylistic bans. |
| $z_{\text{objective}}$ | Implicit optimization objective pursued by the model (e.g., accuracy, creativity, persuasion, or critique). |

We therefore do *not* claim that "11" is minimal or unique. It is an auditable instantiation that supports consistent diagnosis and reporting; depending on the observable proxy family, lower-dimensional subspaces may already be resolution-sufficient, and we report such probe-dependent reductions where appropriate.

**Reporting-resolution check via basis reduction and ablation (relative sufficiency).** To complement the conceptual motivation above, we perform a lightweight *basis reduction* check in a controlled orthogonal design. Because the 11 axes are introduced as an observer-side reporting basis rather than as learned latent factors, the basis-reduction analyses are used as reporting-resolution checks under basis reduction, 11D coarsening, basis rotation, and indicator-set ablation—not as evidence for a unique PCA/factor-analysis optimum or an ontological dimensionality claim. We treat the observer-side coordinate assignment as a fully specified design matrix: each condition is defined by explicit prompt constraints along four dimensions ($z_{\text{epistemic}}, z_{\text{scope}}, z_{\text{format}}, z_{\text{length}}$), encoded as $\pm 1$. We use a balanced $2^4$ design (16 conditions) and sample $n = 50$ independent generations per condition under fixed decoding settings. For each condition, we compute a proxy-delta vector relative to a fixed baseline and fit a simple ridge regression to approximate this proxy-level "signature" from the design coordinates. We then evaluate reduced bases obtained by dropping dimensions (4D→3D→2D→1D). Figure 2 reports mean cross-validated $R^2$ for (i) all proxy targets and (ii) stance-focused targets only, illustrating that the required dimensionality depends on which observables one aims to explain: format/length targets benefit from explicit $z_{\text{format}}$ and $z_{\text{length}}$, whereas stance proxies are largely captured by the $z_{\text{epistemic}}$–$z_{\text{scope}}$ subspace. Target-wise stance results for this orthogonal design are provided in Appendix L.1. For completeness, Appendix L.2 also reports the original 11D reference-basis coarsening check on the cross-lingual politeness probe. Complementary indicator-set ablations on the same generations (10→7→5 observables) are reported in Appendix K, asking whether the redistribution signature persists under coarser observable summaries. Reduced bases (especially 1D–2D) lose descriptive fit for the full proxy family under the same protocol, indicating that redistribution signatures can distribute across multiple observer-side axes rather than collapsing to a single scalar control. This is the only sense in which the present probe family motivates retaining the finer-grained reference basis: not as an intrinsic optimum, but as a useful reporting resolution under the current probe/proxy family.

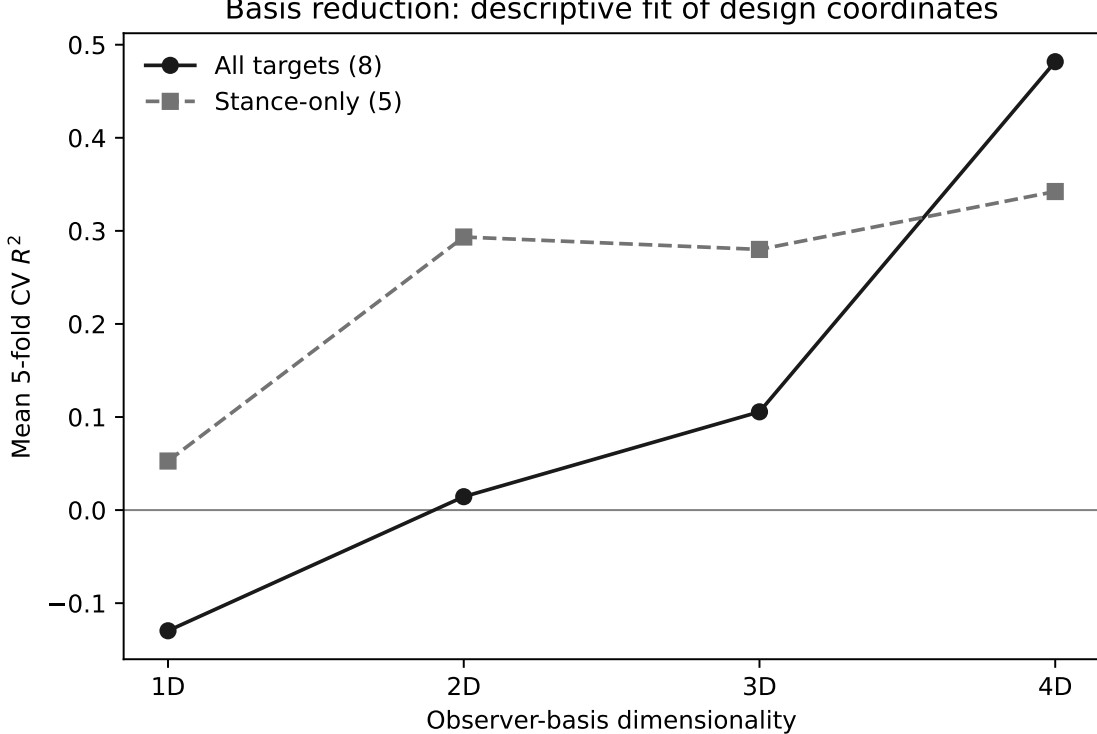

Figure 2: **Reporting-resolution check via basis reduction under an orthogonal 4D design: descriptive fit of design coordinates.** Mean 5-fold cross-validated $R^2$ when approximating each condition's proxy-delta signature from observer-side design coordinates under reduced bases (1D–4D). "All targets" averages over the full proxy set used in the probe (stance + structural/format targets), while "Stance-only" averages over the stance-focused proxies (HEDGE, DEFINITE, COND, ALT_CONTR, ADDITIVE, PROPENSITY). The separation between the two curves highlights that basis sufficiency is probe-dependent: capturing surface realization (format/length) requires explicit $z_{\mathrm{format}}/z_{\mathrm{length}}$, while stance variation is largely explained by the $z_{\mathrm{epistemic}}$–$z_{\mathrm{scope}}$ subspace. This supports *relative* (protocol- and proxy-dependent) descriptive sufficiency rather than a claim of theoretical minimality.

Furthermore, to check that our redistribution detection is not an artifact of a particular axis labeling, we confirm that descriptive fit remains stable under random orthogonal rotations within each macro-group subspace (Appendix L.3; Figure L.3).

**Note on coverage.** While Framing variables (e.g., $z_{\mathrm{audience}}, z_{\mathrm{objective}}$) are structurally important in practice, the main cross-lingual stress test is designed to isolate pathways where typological properties of Japanese make epistemic–scope coupling particularly diagnosable under lightweight observables. Section 3.5 therefore adds a focused supplementary Framing→Reasoning probe using $z_{\mathrm{audience}}$, while a systematic Framing-driven redistribution matrix remains future work (Sec. 5).

### 3.3 Primary and Secondary Effects: Operational Secondary Redistribution

Prompts rarely influence a single observer-side reporting axis in isolation. This decomposition is conceptual rather than algebraic, and does not assume linearity or numerical comparability across dimensions.

For analytical clarity, the observer-side shift associated with a minimal prompt fragment $\pi$ is decomposed as in Eq. (3).

$$\Delta z(\pi) = \Delta z_{\mathrm{primary}}(\pi) + \Delta z_{\mathrm{secondary}}(\pi). \tag{3}$$

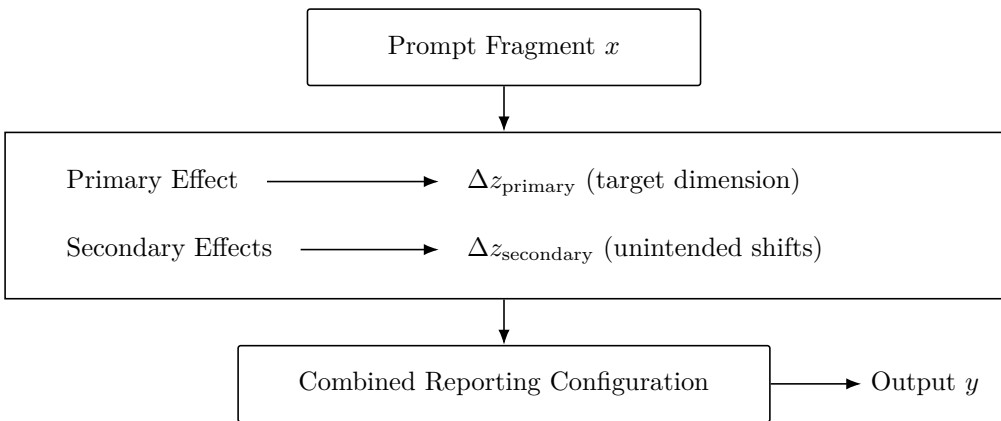

Figure 3: Operational secondary redistribution caused by prompt fragments. A single prompt fragment typically induces a primary shift along its specified reporting axis ($\Delta z_{\text{primary}}$), while simultaneously producing secondary, unintended shifts along other axes ($\Delta z_{\text{secondary}}$).

where $\Delta \boldsymbol{z}_{\text{primary}}(\pi)$ denotes the shift along the dimension most directly specified by the instruction-level semantics of the fragment, and $\Delta \boldsymbol{z}_{\text{secondary}}(\pi)$ captures systematic shifts along other dimensions that co-activate under the same fragment. Fragment-level effects are the focus here because they are easier to isolate and interpret than full-prompt effects.

In this framework, the *primary* dimension of a prompt fragment is determined by its instruction-level semantics, *i.e.*, the most direct operational constraint explicitly imposed on the model. Secondary dimensions are identified empirically as consistent and reproducible changes in observable output proxies induced by the fragment. In cases such as the introductory example, where a fragment primarily encodes an epistemic stance but also induces agreement-seeking closure or scope suppression, these latter effects are treated as secondary rather than as separate primary targets.

Because the same fragment can act as a primary constraint along one reporting axis while inducing structured secondary shifts in others, the Z-model goes beyond a static taxonomy of prompt attributes and supports both forward reporting and post-hoc diagnosis.

*Operational secondary redistribution* is defined as the existence of non-zero secondary effects on observer-side axes other than the intended target. Formally,

$$\exists i \neq j \text{ s.t. } \Delta z_j(\pi) \neq 0, \quad \text{even though the fragment explicitly specifies } z_i. \tag{4}$$

This systematic coupling between primary and secondary prompt effects is referred to as operational secondary redistribution. As illustrated in Figure 3, a single prompt fragment typically induces a primary shift along its target reporting axis, while simultaneously producing secondary shifts along other axes. These secondary effects are not treated as semantic ground truth; they are structured observable consequences of linguistic and pragmatic coupling under a specified protocol.

Throughout this study, the primary dimension of a prompt fragment is defined by its instruction-level semantics (i.e., the most direct operational constraint imposed on the model), while secondary effects are identified empirically via systematic and reproducible changes in output behavior.

Primary effects are often straightforward: "summarize" directly targets $z_{\text{task}}$, while "in bullet points" targets $z_{\text{format}}$. The difficulty is that many Japanese fragments also induce systematic secondary effects. For example, a politeness marker such as 「丁寧に」 primarily increases $z_{\text{style}}$ but can co-vary with $z_{\text{epistemic}}$ in either direction depending on the base framing; cautionary expressions like 「慎重に」 primarily soften $z_{\text{epistemic}}$ but often narrow $z_{\text{scope}}$ and elevate $z_{\text{style}}$; and brevity cues such as 「短く」 primarily reduce $z_{\text{length}}$ while tending to suppress alternatives and increase closure-proneness. The analytical point is not that any one fragment has a

Table 2: Compact examples of prompt fragments and possible secondary shifts. The table is illustrative, not a claim of universal directionality. Japanese examples include romanization and literal English glosses for accessibility; fuller examples are deferred to the appendix.

| Prompt fragment | Literal gloss | Primary target | Possible secondary observable shifts |
|---|---|---|---|
| 「丁寧に説明して」 *teinei ni setsumei shite* | Explain politely | $z_{\text{style}} \uparrow$ | May alter uncertainty, conditionality, or scope markers; this is the pilot pathway tested in Section 3.4. |
| 「慎重に説明して」 *shinchou ni setsumei shite* | Explain cautiously | $z_{\text{epistemic}} \downarrow$ | May increase hedging and conditions while changing register. |
| 「短く答えて」 *mijikaku kotaete* | Answer briefly | $z_{\text{length}} \downarrow$ | May suppress alternatives, caveats, and explanatory scaffolding. |
| 「箇条書きで」 *kajo-gaki de* | In bullet points | $z_{\text{format}}$ | May restructure reasoning presentation and compress scope. |

single universal signature, but that secondary effects are structured enough to be reported, compared, and probed.

Representative prompt fragments and their associated primary and secondary effects are summarized in Table 2. Appendix H provides a compact worked example of how the same vocabulary can be used as a post-hoc diagnostic scaffold. The Z-model is a diagnostic vocabulary for explaining prompt engineering as interaction-level configuration of an observable behavioral distribution, rather than as surface-level linguistic manipulation or model-internal control.

**Operationalization (observable secondary redistribution).** While $\mathbf{z}$ denotes a conceptual observer-side configuration, all empirical claims in this work are grounded in *observable* output-level measurements. Specifically, we operate on a vector of proxy features $r(y)$ computed from model outputs, whose full definitions and extraction procedures are provided in Appendix G.[1]

We say that a prompt perturbation $\pi$ exhibits *operational secondary redistribution* if a fragment intended to primarily target reporting axis $i$ induces a reproducible and directionally consistent change in at least one proxy feature associated with a distinct axis $j \neq i$. Reproducibility is assessed under matched prompts, fixed decoding parameters, and distribution-level comparisons across repeated runs (Section 3.4).

This definition is deliberately agnostic to the underlying model internals: redistribution is established solely through observable cross-axis effects in $r(y)$, without assuming direct access to internal states.

### 3.4 Pilot Japanese/English politeness probe under matched tasks

The empirical study is a pilot diagnostic case study of one pathway: an Expression-oriented politeness cue and its secondary effects on epistemic- and scope-related surface markers. Japanese/English is used as a diagnostic setting because the same high-level cue can be realized by different pragmatic and morphosyntactic resources; we do not interpret absolute rates across languages, and we do not claim a clean typological causal effect.

Five benign technical topics are used, chosen to admit limitations, conditions, and alternative explanations without requiring disputed factual knowledge. For each topic, language, and style condition, we sample $n = 50$ independent generations; aggregating across five topics yields $N = 250$ outputs per language and style. All main API-served generations used the pinned OpenAI API snapshot `gpt-5.2-2025-12-11` (reported alias: GPT-5.2) during the 2026-02-27 UTC execution window; the bundle records run-level provenance for this API-served run. For each language, the task description, system prompt, decoding parameters, and output constraints are held constant; the user-level fragment is either present ("politely" / 「丁寧に」, *teinei*

---

[1]Each proxy is defined independently of any single experiment and is reused across all evaluations in Section 3.4.

*ni*, "politely") or absent. The purpose of the experiment is not to score correctness. It is to ask whether a surface register cue is accompanied by directional redistribution in observable discourse markers.

**Proxy definitions and revision-specified primary endpoints.** We use transparent lexical and structural features after deterministic Unicode normalization (NFKC). For the revised analysis, after narrowing the empirical claim and before computing the revised primary tables and figures, we fixed four primary endpoints: epistemic hedges (HEDGE_EPI), generality/typicality hedges (HEDGE_GEN), conditional markers (COND), and contrastive-alternative markers (ALT_CONTR). ALT_CONTR is retained symmetrically across Japanese and English because it was part of the revision-specified endpoint family after separating contrastive alternatives from additive discourse markers; in the present English politeness protocol it is sparse and has near-zero variance, so EN ALT_CONTR entries are interpreted as protocol-level nonactivation rather than as a substantive semantic null effect. This is a post-collection revision specification, not a preregistration or a claim of blindness to the raw generations; all other lexical, structural, reader, and correlation analyses are treated as exploratory or descriptive. In response to the ambiguity of the earlier ALT proxy, we split broad scope-expansion markers into (i) *additive* discourse markers such as *also/furthermore* and また/さらに, and (ii) *contrastive-alternative* markers such as *alternatively/on the other hand* and 一方で/別の可能性. Only ALT_CONTR is treated as a primary alternative-reasoning proxy; ADDITIVE is reported as exploratory. All proxies are surface markers and are not calibrated semantic measurements of uncertainty, reliability, or safety.

**Measurement note: enumerative breadth versus perceived sufficiency.** The most important measurement caveat concerns the construct of "coverage" or "scope." Automatic discourse proxies are sensitive to *enumerative breadth*: more listed conditions, more contrastive alternatives, more additive markers, or longer explanations. Human readers may instead evaluate coverage as *relevance-weighted sufficiency*: whether the necessary points are present, non-redundant, and appropriately developed for the question. We therefore do not expect lexical scope proxies and reader-rated scope/sufficiency to coincide perfectly. In this paper, lexical indicators are interpreted as surface-marker redistribution, while reader ratings are reported descriptively to show whether such redistribution is mirrored by a small set of external-reader judgments about breadth, sufficiency, or scaffolding. The construct distinction is grounded in the measurement literature, not inferred from noisy reader correlations.

**Statistical protocol.** The original output-level bootstrap is replaced in the main analysis by a topic-aware hierarchical bootstrap. Each bootstrap replicate resamples topics with replacement and then resamples generations within each selected topic-condition cell. This reflects the nested design (topic → generation) and generally yields wider intervals than treating 500 generations per language as exchangeable observations. For the four primary endpoints in two languages, we report Benjamini–Hochberg FDR correction across the eight tests. Exploratory metrics, the reader endpoints, the supplementary Exp. C probe, and the proxy–reader correlation tables are reported separately and are not used as primary evidence.

**Main pilot findings.** After topic-aware resampling and FDR correction, the robust primary findings are localized. In English, the politeness cue increases epistemic hedging (HEDGE_EPI: $\Delta = 0.237$ per 1,000 characters, $q = 0.003$). In Japanese, the politeness cue increases conditional framing (COND: $\Delta = 0.826$ per 1,000 characters, $q = 0.016$). The same two findings remain positive under raw-count checks (EN HEDGE_EPI: +0.236 [0.076, 0.428], raw-count $q = 0.011$; JP COND: +0.932 [0.508, 1.372], raw-count $q = 0.010$), which addresses the concern that per-character normalization alone could create the result. Other primary shifts should be read more cautiously: Japanese HEDGE_GEN is directionally negative but its topic-aware CI crosses zero, and Japanese ALT_CONTR is directionally negative with a CI excluding zero but does not pass the primary FDR threshold ($q = 0.079$). Detailed numerical tables, including effect sizes and exploratory structural endpoints, are moved to Appendix D; the main text emphasizes the visual forest plots and the two localized robust API-window effects. The two automatic findings have different robustness profiles: EN HEDGE_EPI is the more stable cross-model signal, whereas JP COND is robust within the API-served politeness probe and raw-count check but is not present on the pinned open-weight sensitivity comparison (Appendix J). Thus, the pilot warrants only a narrower conclusion than the earlier version: a

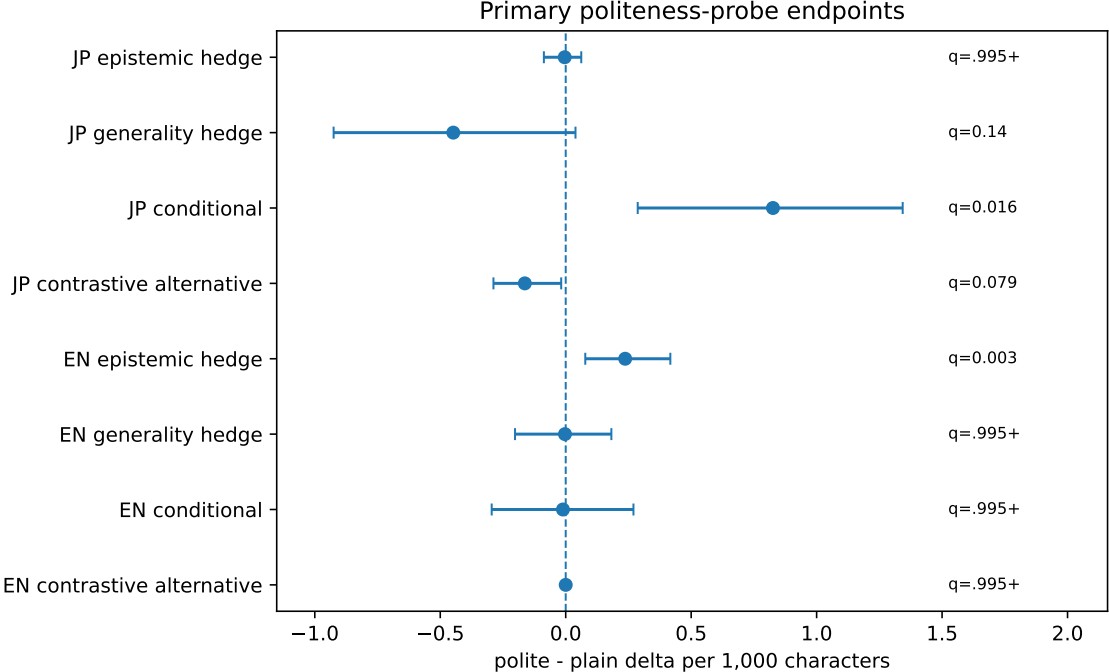

Figure 4: Primary politeness-probe endpoints under topic-aware hierarchical bootstrap. Points show polite–plain deltas per 1,000 characters, with 95% CIs and Benjamini–Hochberg adjustment across the eight revision-specified primary tests. Takeaway: the robust effects are localized to English epistemic hedging and Japanese conditional framing; other endpoints are interpreted as directional or exploratory rather than as a global redistribution signature.

style cue can be accompanied by measurable secondary marker redistribution, but the effects are pathway-, proxy-, model-, and snapshot-dependent rather than a broad validation of the 11-axis framework.

**Open-weight model-sensitivity comparison.** A pinned open-weight comparison on Qwen/Qwen2.5-7B-Instruct is retained in Appendix J as a snapshot-stable sensitivity path. The comparison is deliberately not summarized as broad cross-model validation: it shows the English hedging direction most clearly, but the Japanese COND increase that is robust in the API-served pilot is near zero on the pinned open-weight checkpoint. We therefore treat the open-weight result as evidence about model/snapshot sensitivity of the reporting protocol, not as an independent model-family validation. Indicator-set ablations are likewise moved to Appendix K and are used as robustness checks for the reporting protocol, not as evidence that the proxy family captures all semantic aspects of uncertainty or scope.

**Language-qualified direct-reader check.** To complement the automatic markers, we added a condition-blind direct-reader check on the same balanced subset of 200 outputs (JP/EN × plain/polite; 50 outputs per cell). Each output was rated on five-point scales for categorical tone (Q1), uncertainty/qualification (Q2), scope/sufficiency (Q3), and explanatory scaffolding (Q4), without prompt-condition labels. We analyze the reader judgments by *direct-reading eligibility* for each language. The inclusion rule for the reported reader analyses is based only on whether the annotator could evaluate the corresponding language output directly, without machine translation; it is not based on the direction or magnitude of the ratings. Two multilingual readers directly rated both Japanese and English outputs. Two additional Japanese-proficient readers directly rated the Japanese outputs; their English judgments may have involved machine translation and are therefore not pooled with the main English direct-reader check. This avoids validating English prompt effects through a translation layer that can itself alter hedging, conditionality, and discourse-scope cues. The ratings are reported as exploratory interpretive context rather than as a decisive validation or invalidation instrument.

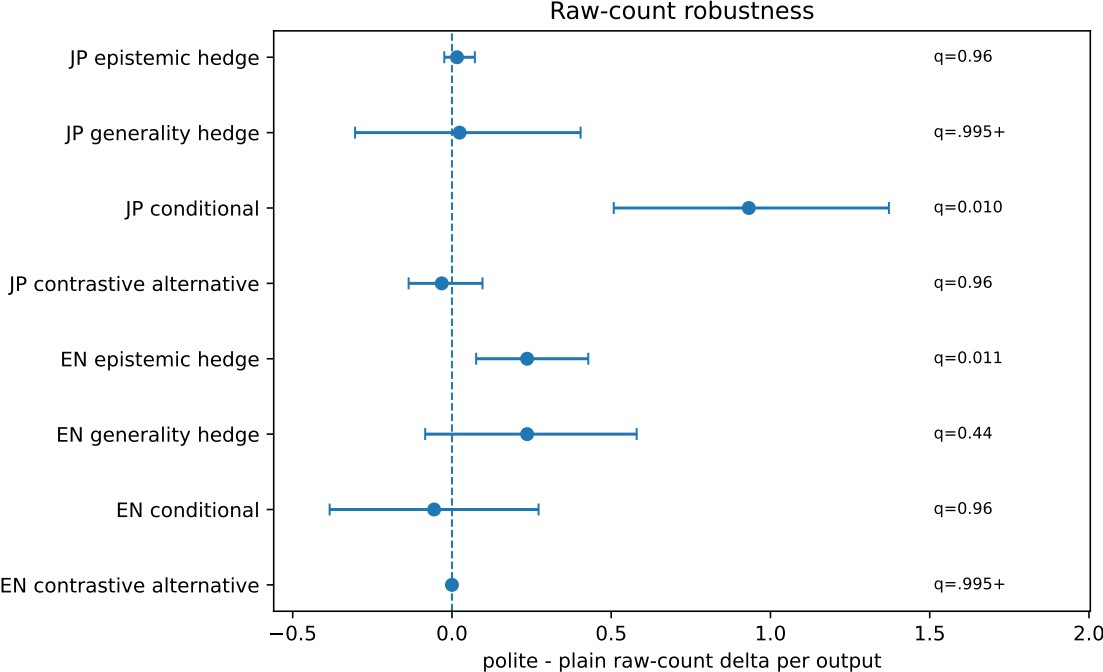

Figure 5: Raw-count robustness for the same revision-specified politeness-probe endpoints. Points show polite–plain raw-count deltas per output, with topic-aware hierarchical bootstrap 95% CIs and Benjamini–Hochberg adjustment across the eight raw-count checks. Takeaway: the two rate-based primary findings, English HEDGE_EPI and Japanese COND, remain positive under raw counts, reducing the concern that they are artifacts of character-length normalization.

Eligibility and agreement diagnostics are reported in Appendix E; pairwise QWK values are near zero, so the reader check is not treated as a reliable semantic measurement instrument. The reader endpoints and proxy–reader correlations are reported for transparency rather than used to establish, confirm, or rule out semantic effects.

The language-qualified reader check gives mixed exploratory context. Because agreement diagnostics are low (Appendix E), we do not use the reader study to validate, invalidate, or adjudicate the lexical proxies. English direct-reader ratings move upward for uncertainty/qualification and scope/sufficiency under polite prompts; because the English direct-reader set has only two raters and low agreement, we treat this as directional, hypothesis-generating evidence only and do not rely on adjusted $q$ values from the reader table as inferential support. Japanese direct-reader ratings do not show robust perceived shifts in categorical tone, uncertainty, or scope/sufficiency; the only consistent movement is a directional increase in explanatory scaffolding. We therefore keep the Japanese automatic COND increase at the level of conditional/explanatory surface-marker redistribution, not human-validated pragmatic broadening. Appendix E also reports condition-stratified proxy–reader Spearman correlations. These correlations are descriptive and mixed; because reader agreement is low and several same-construct pairings are weak or negative, we do not interpret low, null, or negative correlations as evidence for the construct distinction. The distinction between enumerative marker quantity and reader-perceived sufficiency is a measurement assumption grounded in prior construct-validity work, and the correlation table is included only to make the proxy–perception relation transparent.

**Exploratory factorial check.** We retain a small $2 \times 2$ factorial probe on one representative topic as an exploratory direction-to-observation check. It crosses the politeness cue with an anti-closure instruction and asks whether the combined condition deviates from additivity. Because the probe uses one representative topic and two prompt-task templates, with 30 generations per task–condition cell and the reported ALL

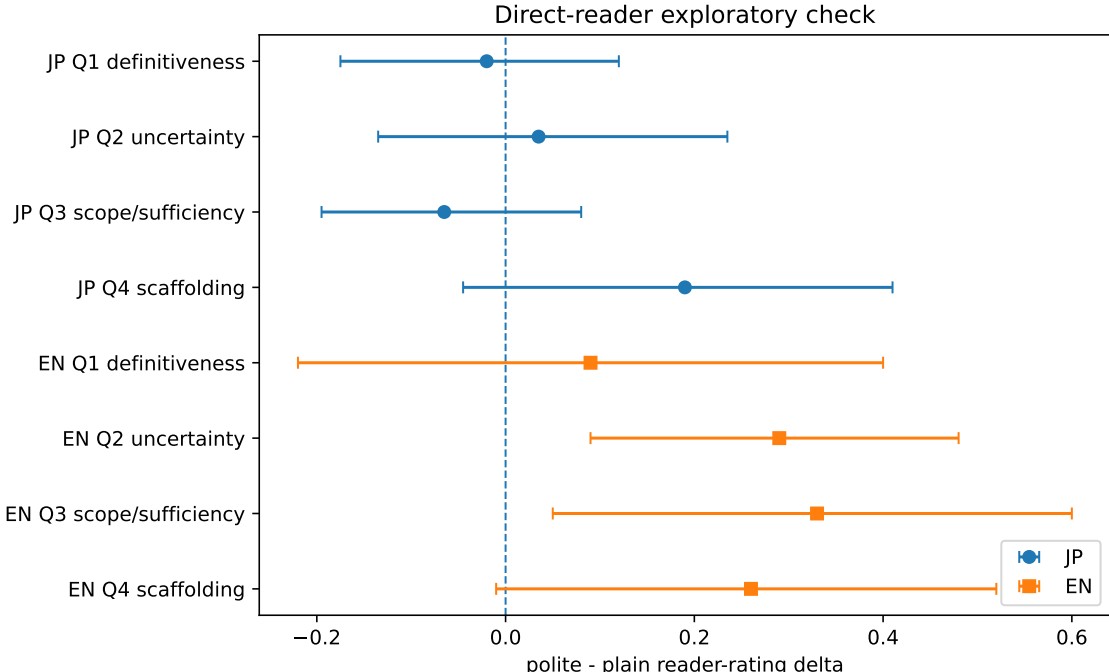

Figure 6: Language-qualified direct-reader exploratory check. Points show polite–plain deltas on the 1–5 reader scale, with topic-aware hierarchical bootstrap 95% CIs. We omit multiplicity labels from the plot because the reader check is small and exploratory; Table E.2 reports the corresponding adjusted values only for transparency. Q1=definitiveness, Q2=uncertainty/qualification, Q3=scope/sufficiency, and Q4=explanatory scaffolding. Circles indicate Japanese direct-reader ratings ($n = 4$); squares indicate English direct-reader ratings ($n = 2$). Takeaway: because inter-reader agreement is low, the reader check is reported as descriptive context, not as confirmatory validation or a decisive semantic test. English Q2/Q3 move in the expected direction but remain hypothesis-generating; Japanese Q3 does not support upgrading the JP marker shift into a perceived-scope claim.

slice pooling both templates ($n = 60$ outputs per condition), it is not used as primary evidence; the main manuscript no longer summarizes it in a dense table, and detailed interaction plots and cell means are reported in Appendix L.4.

## 3.5 Supplementary Framing→Reasoning probe: audience framing as a second audited path

To show that the macro-group vocabulary can be instantiated beyond the main politeness pathway, we add a compact supplementary audience-framing probe in the main text. The manipulation targets $z_{\text{audience}}$: a neutral explanation prompt is contrasted with a matched prompt for a non-specialist or novice reader (English: "for a non-specialist reader"; Japanese:「初学者にもわかるように」, *shogakusha ni mo wakaru you ni*, "so that even a beginner can understand"). The design matches Exp. A in size ($N = 250$ outputs per language and audience condition) and keeps the task, topics, model settings, and output constraints fixed. Because Exp. C was executed in a later API window than the main politeness probe, it is interpreted as a second audited pathway under the same protocol family, not as same-snapshot joint evidence or as evidence that a single model state supports all pathways simultaneously.

This probe is a support layer around the main politeness study rather than a second primary endpoint family. Using the raw 5-topic × 2-language × 2-audience-condition × 50-generation dataset, the novice/non-specialist cue increases analogy-oriented and explanatory-scaffolding markers in both English and Japanese. The effect is not a uniform increase in all reasoning-related proxies: conditional markers do not increase consistently.

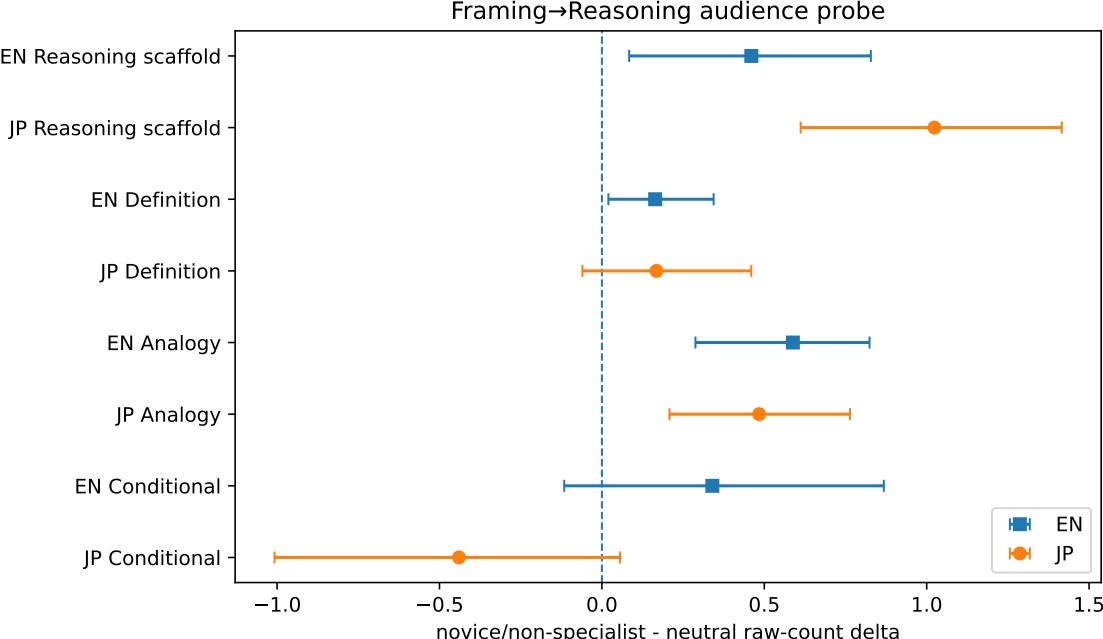

Figure 7: Supplementary Exp. C Framing→Reasoning audience-framing probe, recomputed from raw generations. Points show novice/non-specialist audience minus neutral raw-count deltas per output, with topic-aware bootstrap 95% CIs. Squares denote English; circles denote Japanese. Takeaway: the audience cue increases analogy-oriented and explanatory-scaffolding markers in both languages, while conditional markers are not uniformly increased. This is a second audited path and remains a supplementary support layer rather than a second primary endpoint family.

We therefore interpret Exp. C as evidence that the macro-group vocabulary can audit a second localized redistribution pathway, not as broad validation of Framing or Reasoning as full validated constructs. The full prompt templates, dictionary, raw-count tables, stricter analogy check, and data-quality report are reported in Appendix F.

## 3.6 Reproducibility note

To isolate prompt-induced effects, we fixed the base prompt, system prompt, and maximum output length, and varied only the probe fragment, while holding decoding parameters constant. The quantities needed for protocol re-execution are therefore the prompt templates, model identifiers (and, for API-served models, the execution window), sampling settings, proxy definitions, and post-processing rules. These are specified explicitly in the appendix rather than left to an external code release.

For the main cross-lingual probe (Exp. A), we sample $n = 50$ generations per topic and condition; with five topics this corresponds to $N = 250$ outputs per language and style. For the supplementary audience-framing probe (Exp. C; Section 3.5), we use the same $n = 50$ per topic and condition design, yielding $N = 250$ outputs per language and audience condition. In this revision, the Exp. C figure and appendix tables are recomputed from the raw 1000-generation JSONL rather than from hand-entered summary values. The intended re-execution target is recovery of the reported *directional, distribution-level deltas* under the matched protocol, not bitwise string identity.

Readers who prefer a snapshot-stable comparison path can run the same prompts and counting rules on the pinned open-weight checkpoint reported in Appendix J.

Table 3: Claim–evidence boundary for the revised pilot. The table states the level at which the evidence is used and the claims that are explicitly outside the paper's scope.

| Evidence layer | Claim made in this paper | Not claimed |
|---|---|---|
| Automatic regex and structural proxies | Transparent, deterministic detection of surface-marker redistribution under matched perturbations. | Estimation of reader-perceived semantics, factuality, safety, or calibrated uncertainty. |
| Topic set | Controlled within-protocol differences on five benign technical topics. | Generalization to high-risk domains, adversarial prompts, or dialogue tasks in general. |
| Open-weight comparison | Model/snapshot-sensitivity diagnosis for the reporting protocol. | Cross-model linguistic or pragmatic universality of the Japanese COND effect. |
| 11-axis Z-model | A common audit scaffold for naming primary targets and secondary observable marker families. | Recovery of hidden model states, proof of a unique continuous latent space, or empirical validation of the full 11-axis theory. |
| Reader and LLM-judge checks | Boundary diagnostics showing where marker evidence should stop. | Confirmatory semantic validation of the automatic proxies. |

The extended Japanese linguistic background is moved to Appendix A. In the main text, these mechanisms are used only to motivate the Japanese/English diagnostic setting and the choice of a politeness cue; they are not treated as evidence that the observed effects are typological or general across models.

## 4 Analysis and Discussion

### 4.1 What the pilot supports

The revised evidence supports a localized, protocol-level claim. A register cue can be audited for secondary movement in observable discourse markers under matched black-box perturbations. For the main politeness probe, the strongest automatic findings are EN HEDGE_EPI and JP COND, both surviving topic-aware hierarchical bootstrap, FDR handling, and raw-count robustness checks. The supplementary audience-framing probe adds a raw-data-derived second audited path, Framing→Reasoning, but it is presented as a support layer rather than a new primary endpoint family. The reader check is intentionally non-confirmatory: it reports whether marker movement is mirrored by a small set of external readers, but it does not establish that marker counts are calibrated measures of perceived semantics.

This is still useful for auditing. Marker redistribution can be valuable even when it is not perceived by external readers as a broad semantic change: it can reveal regression across model snapshots, detect prompt variants that unexpectedly increase conditional or hedging markers, and provide a cheap monitoring layer before expensive semantic review. The claim is therefore not that markers equal meaning; it is that a transparent marker layer can make selected off-target prompt effects observable, comparable, and reproducible.

### 4.2 Construct validity: automatic surface proxies versus reader-perceived sufficiency

The near-zero inter-reader agreement in the exploratory check is not treated as a failed attempt to validate the regex layer as a semantic instrument. It is a boundary diagnostic. The automatic proxies are defined and interpreted only as a cheap, transparent monitoring layer for deterministic surface-marker redistribution under matched perturbations. They do not, and are not intended to, serve as estimators of reader-perceived semantics, relevance-weighted sufficiency, or pragmatic broadening. This boundary is central to the paper's evidence-matching strategy: marker movement can support an audit claim, but it cannot by itself support a claim about what readers semantically perceive.

The reader study highlights a construct-validity issue rather than merely a failed validation check. In measurement terms, *reader-perceived sufficiency* is the target construct, while marker counts such as COND, ALT_CONTR, ADDITIVE, sentence counts, or list items are operationalizations. Measurement-modeling work emphasizes that unobserved constructs must be inferred from observable properties, and that construct–

operationalization mismatches can produce misleading conclusions if left implicit (Jacobs & Wallach, 2021). NLP measurement critiques make the same point in language-technology settings: quantitative proxies should be tied explicitly to the construct they purport to measure (Blodgett et al., 2020).

We therefore separate two notions of scope. *Enumerative breadth* denotes observable quantity: more conditions, alternatives, discourse markers, or explanatory pieces. *Reader-perceived relevance-weighted sufficiency* denotes whether a reader judges that the necessary points are present, non-redundant, and appropriate for the question. This distinction explains why JP COND can be a robust automatic finding while JP Q3 scope/sufficiency does not increase in the four-reader Japanese direct-reader check. The reader check is useful here as exploratory context: because reader judgments do not uniformly mirror marker movement, we do not upgrade marker quantity into perceived-coverage claims.

### 4.3 What the Z-model adds beyond a prompt-feature taxonomy

A simpler taxonomy can label a prompt fragment as "polite," "brief," or "for novices." The Z-model adds a reporting convention for *primary versus secondary* effects: the prompt fragment identifies an intended target axis, while the paired probe asks which off-target marker families move under the same protocol. This distinction makes results comparable across perturbations: Expression→Epistemic/Scope for the politeness probe and Framing→Reasoning for the audience-framing probe. The framework is therefore useful as an audit scaffold even without estimating hidden states.

The forward/inverse terminology is likewise descriptive. Forward use means specifying a fragment, its intended primary target, and a small set of observable secondary indicators before sampling. Inverse use means using output-level indicators after generation to organize a post-hoc explanation of which observer-side axes appear to have shifted. Neither direction is a unique hidden-state recovery procedure. Expanded worked examples remain in Appendix H; the main text keeps only the minimal logic needed to interpret the empirical probes.

### 4.4 Japanese and English roles in the revised design

Japanese remains a diagnostic setting because politeness, stance, and sentence-final modality can be expressed compactly at the surface. However, the revised interpretation no longer treats Japanese as typological proof. For Japanese, the robust automatic result is marker-level: the polite cue increases conditional/explanatory framing, while the direct-reader check does not support a robust perceived-scope increase. For English, the direct-reader subset provides only directional, hypothesis-generating signals for uncertainty/qualification and scope/sufficiency; because $n = 2$ and agreement is low, these ratings are not used to establish semantic effects. This division keeps the language story aligned with the evidence: Japanese is the clearest marker-redistribution case, while English provides the clearest but still exploratory reader-perception signal.

### 4.5 Practical interpretation

The practical outcome is an audit recipe rather than a prompt-control method. First, state the intended primary target of a fragment. Second, define a small revision-specified set of off-target marker families. Third, estimate distribution-level deltas with topic-aware uncertainty and multiple-comparison handling. Fourth, add robustness and perception-context checks before upgrading marker movement into semantic claims: raw counts for length artifacts, direct-reader context for perceived semantics, and stricter dictionaries for broad lexical categories. Finally, report negative or inconclusive checks as part of the interpretation rather than using them as decisive validation or invalidation. This recipe is the main methodological contribution of the revised paper.

## Reproducibility Statement

This paper studies a *protocol-level* phenomenon rather than introducing a trainable model, a new benchmark, or a large derived dataset. Accordingly, the central reproducibility requirement is exact reporting of the interaction protocol: (i) prompt templates and probe fragments, (ii) model identifiers and, when relevant, pinned API snapshot strings and execution windows, (iii) decoding settings, (iv) observable proxy definitions

and normalization rules, and (v) aggregation and uncertainty-estimation procedures. We include each of these elements in the manuscript and appendices. The supplementary artifact bundle for this revision further includes anonymized raw generations, prompt metadata, response-level proxy features, the revised ALT/ADDITIVE split, the topic-aware bootstrap script, generated CSV summaries, the condition-blind reader-check form, and language-qualified reader-check summaries, including the Japanese direct-reader and English direct-reader/sensitivity split. One additional completed reader form with uncertain direct-reading eligibility is excluded from all paper analyses; the bundle retains only a note documenting this exclusion and does not include the raw unverified form. The bundle also includes the rubric-based LLM-judge requests, generated judge ratings, condition-specific proxy–judge correlation summaries, and analysis scripts for the main politeness-probe outputs. These judge results are included only as descriptive semantic-check diagnostics and are not used as a substitute for the reported reader study. For the supplementary Framing→Reasoning probe, the bundle includes the raw audience-framing generations (5 topics × 2 languages × 2 audience conditions × 50 generations), response-level reasoning-presentation proxy counts, topic-aware bootstrap summaries, the data-quality report, and the plotting script used for the Exp. C dot plot. The supplementary zip includes sanitized raw generations, prompts, derived features, run-level API provenance metadata, and request/response artifacts where retained; credentials and organization-specific identifiers are excluded. The small reader annotation should also be reflected consistently in the OpenReview/TMLR human-subjects reporting metadata, according to the authors' institutional or company policy; the manuscript does not rely on personal or sensitive information about the readers.

A reader can re-execute the protocol from the paper and supplementary artifacts. Appendix C specifies the prompt construction, probe fragments, decoding conditions, and sampling design used in the minimal probes; Appendix F provides the supplementary audience-framing probe. Appendix G lists the observable indicators and the regex-based dictionary used to operationalize them after deterministic Unicode normalization (NFKC). Appendix I summarizes the API settings, reporting template, and reproducibility checklist. No fine-tuning, hidden retrieval corpus, or learned post-processing model is involved in the automatic proxy measurements. The direct-reader annotation reported in Section 3.4 is a separate reader-check layer and is released as supplementary evaluation material.

For the main API-based experiments, all generations were obtained via the OpenAI API using the pinned OpenAI API snapshot `gpt-5.2-2025-12-11` (reported alias: GPT-5.2) during the execution window 2026-02-27 UTC. The system prompt was fixed across conditions, and only the user-level probe fragment was varied. Unless otherwise stated, decoding parameters were held constant (temperature $= 0.7$, top_p $= 1.0$, maximum output length $= 512$ tokens), with $n = 50$ independent generations per condition. The supplementary audience-framing probe (Exp. C) used the same pinned OpenAI API snapshot `gpt-5.2-2025-12-11` and decoding settings during 2026-05-23–24 UTC. The artifact bundle stores its 1000 raw audience-framing generations under `data/exp_c_framing_reasoning/raw/runs.jsonl` and records run-level provenance under `data/exp_c_framing_reasoning/raw/run_metadata.json`. Because API-served models can drift over time and stochastic sampling is not guaranteed to be bitwise deterministic, the intended protocol re-execution target is *directional and distributional agreement* under the same protocol rather than exact string identity. Concretely, a successful protocol re-execution should recover the sign and qualitative ordering of the main reported deltas, while allowing magnitudes to vary across snapshots or model families.

To reduce dependence on a single API snapshot, Appendix J reports a second model-sensitivity comparison path on a pinned open-weight checkpoint: `Qwen/Qwen2.5-7B-Instruct`, revision `a09a35458c702b33eeacc3 93d103063234e8bc28`. This open-weight comparison uses matched prompts, decoding settings, and proxy definitions, providing a stable reference point for readers who prefer a non-API comparison path.

## 5  Limitations

The study remains a pilot diagnostic case study. It tests two localized pathways—Expression→Epistemic/Scope in the politeness probe and a supplementary Framing→Reasoning audience probe—rather than a full redistribution matrix over the 11-axis reference basis. The API-served model may drift over time, the open-weight sensitivity comparison is only one additional checkpoint, and the topic set is deliberately small

and benign. Accordingly, the relevant protocol re-execution target is directional, distribution-level agreement under the matched protocol, not exact strings or universal effect sizes.

The proxy layer is intentionally lightweight. Regex and structural features are useful for transparent monitoring, but they are not calibrated measures of factuality, safety, or reader-perceived sufficiency. This is why the revision adds raw-count checks, splits ADDITIVE from ALT_CONTR, and treats DEFINITE as exploratory. The language-qualified reader study is also limited: four readers directly rated Japanese outputs, but only two readers directly rated English outputs; agreement is low enough that the reader study is reported as exploratory context, not as confirmatory semantic validation or a decisive test. Native-speaker Japanese pragmatic annotation and additional direct-English readers remain important next steps. If the goal is to upgrade the marker-level audit into a calibrated semantic-perception framework, the appropriate roadmap is a larger direct-reader pool, structured pairwise comparison protocols, and explicit annotator calibration sessions; that would be a different evaluation layer from the present lightweight marker audit.

The work should not be read as a prompting algorithm, safety certification method, or estimator of hidden model states. It provides a reporting vocabulary and a reproducible audit protocol for selected off-target prompt effects. Broader task families, adversarial prompts, high-stakes domains, more model families, richer semantic annotation, and direct tests of downstream user decisions remain future work.

## 6 Conclusion

This paper introduced the Z-model as an observer-side vocabulary for reporting prompt-induced behavioral shifts under black-box access. The framework is intentionally descriptive: it is a reporting vocabulary for prompt–response behavior, not a latent-state estimator or prompt-control algorithm. The empirical contribution was narrowed to a pilot Japanese/English politeness probe of one Expression→Epistemic/Scope pathway. With topic-aware hierarchical bootstrap and revision-specified primary endpoints, the strongest automatic-proxy effects are localized: English politeness increases epistemic hedging, and Japanese politeness increases conditional framing. Other marker shifts are reported as directional or exploratory, especially after splitting additive discourse markers from contrastive-alternative markers. The language-qualified reader check keeps the Japanese interpretation at the marker-redistribution and explanatory-scaffolding level, and it treats English uncertainty/qualification and scope/sufficiency movements as directional signals rather than established semantic effects.

The revised results support a precise claim: local register cues can be audited for sparse secondary surface-marker redistribution under a matched protocol. The supplementary audience-framing probe adds a raw-data-derived second audited path, showing that the macro-group vocabulary can also organize localized Framing→Reasoning effects such as analogy-oriented and explanatory-scaffolding marker movement. Together, these probes do not make the 11-axis reference basis a validated latent estimator, but they demonstrate how a scoped vocabulary can specify which prompt fragment was perturbed, which observable markers moved, which checks support the interpretation, and which claims should remain outside the evidence.

The practical value of this protocol is not that any single marker is universal, but that prompt-induced marker redistribution can be audited reproducibly across model snapshots, prompt variants, and languages. In this sense, the vocabulary functions as a lightweight regression-monitoring layer: it can flag when a local prompt edit changes epistemic or scope-related presentation, when a marker shift is stable across model families, and when it is model- or snapshot-dependent. Such distinctions are useful precisely because they prevent local prompt effects from being mistaken for general semantic or reliability properties.

Future work should add native-speaker Japanese pragmatic annotation, additional direct-English readers, broader fragment/topic/model-family coverage, and richer semantic annotations that distinguish enumerative breadth from relevance-weighted sufficiency. A natural next step, if the target becomes semantic-perception calibration rather than marker auditing, is to replace absolute Likert ratings with structured pairwise comparisons and calibration sessions; this remains outside the evidence-bearing claim of the present pilot.

## 7  Broader Impact Statement

This study proposes an observer-side diagnostic vocabulary (the Z-model) for diagnosing how small prompt variations can induce coupled shifts in style, epistemic stance, scope, and reasoning behavior in black-box LLM interactions. A potential negative impact is that the same mechanisms can be misused to elicit overly definitive or authoritative-sounding outputs, increasing the risk of over-trust, premature closure, or misinformation in downstream decision-making. These risks are particularly relevant in high-stakes settings where users may interpret polite or confident language as reliability. To mitigate such harms, our goal is explicitly diagnostic rather than prescriptive: we emphasize lightweight A/B probing, transparent surface-level indicators, and prompt patterns that encourage conditionalization and explicit uncertainty when appropriate. The experiments use benign technical topics and report relative directional shifts under matched conditions rather than claiming general correctness or deployment readiness. These observations should be read as *diagnostic implications*: they suggest where prompt cues may redistribute stance- and scope-related behaviors under a given protocol, but they do not imply a method to control, certify, or guarantee safe outcomes. We encourage practitioners to apply the framework as a safety-oriented auditing tool, and to combine it with human oversight and domain-specific verification in real applications.

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

# APPENDIX

Supplementary and Reproducibility Details

## A    Japanese as a Diagnostic Setting

This appendix collects the linguistic motivation that was previously in the main text. The mechanisms below motivate why Japanese can be useful for a compact pilot probe, but they do not by themselves establish typological generality or model-independent causal mechanisms.

Table A.1 summarizes Japanese linguistic mechanisms that motivate the diagnostic setting. These are hypotheses and background considerations, not conclusions established by the pilot probe.

**Interpretive status.**    The table is used as background for designing probe families. The empirical claims in the main text remain limited to the revision-specified surface-marker endpoints and the matched protocol in Section 3.4.

## B    Interaction-Level Epistemic Control

This appendix collects concrete prompt templates, a minimal probing protocol, and language-specific observations that instantiate the interaction-level epistemic control discussed in the main text. The intent is descriptive and operational: to make prompt-induced epistemic effects easy to reproduce and observe, not to prescribe a normative style guide.

For reference, Table B.1 provides a compact reference sheet of formal definitions for the eleven observer-side reporting axes in the Z-model.

### B.1    Minimal Reliability-Oriented Prompt Templates

We first present a compact set of prompt templates intended to encourage reliable, non-authoritative responses without requiring domain expertise. These templates primarily modulate epistemic stance and scope licensing rather than changing propositional content.

**Prompt A (Epistemic softening).**

> **JP:** X について、基本の見取り図を知りたいです。
> 断定は不要で、成り立つ条件や怪しい条件があれば教えてください。
> **EN:** I would like a basic overview of X. Definitive answers are not required; please describe the conditions under which it holds and cases where it may be questionable.

**Prompt B (Default framing with permission to expand).**

> **JP:** X は基本的には Y で考えていいんでしたっけ？
> もし他にも押さえるべき観点があれば、代表例だけ教えてください。
> **EN:** Is it generally acceptable to understand X in terms of Y? If there are other important aspects to consider, please mention representative ones.

**Prompt C (Evidence–inference separation; RAG-compatible).**

Table A.1: Japanese linguistic mechanisms motivating possible prompt–LLM coupling. We organize prior observations into **four background mechanisms** consistent with the primary/secondary-shift schematic in Figure 3; additional examples (e.g., honorific morphology, flexible word order) are included as representative sub-phenomena under the relevant mechanism. Arrows denote typical tendencies relative to a neutral baseline; *drift*/$\pm$ indicate context-dependent direction; $\rightsquigarrow$ denotes qualitative restructuring.

| Core mechanism | Representative sub-phenomena (JP) | Why it may couple dimensions | Typical z-effects (primary $\rightarrow$ secondary) |
|---|---|---|---|
| Politeness–epistemic coupling | 丁寧語・敬語, です／ます調, 尊敬語・謙譲語 (honorific morphology) | Politeness and honorific marking encode interactional alignment and social stance compactly. In Japanese, these cues can be co-interpreted as epistemic authority or responsibility signals, so a stylistic instruction can co-activate epistemic stance (the sign may depend on base framing). | $z_{\text{style}} \uparrow \rightarrow$ $z_{\text{epistemic}}(\uparrow / \downarrow)$, $z_{\text{process}} \uparrow$ |
| Argument omission and role ambiguity | 主語／目的語の省略 (ゼロ代名詞), 話者／聞き手の省略 | Omission increases the inference burden for recovering roles and addressees from context. This can induce instability in inferred persona and audience, and propagate into reasoning alignment and stance. | $z_{\text{role}}(\text{drift})$, $z_{\text{audience}}(\text{drift}) \rightarrow$ $z_{\text{epistemic}}(\pm)$, $z_{\text{process}}(\pm)$ |
| Vague modifiers and pragmatic overloading | 慎重に, 丁寧に, 適切に, わかりやすく, なるべく… | Semantically underspecified but pragmatically rich modifiers bundle multiple implied constraints (e.g., tone, caution, coverage, and reasoning strategy), making secondary marker shifts plausible under some prompt regimes. | $(\textit{varies}) \rightarrow$ $z_{\text{style}} \uparrow$, $z_{\text{epistemic}} \downarrow$, $z_{\text{scope}} \downarrow$ |
| Topic prominence and discourse-driven structure | は (topic marking), 主題化, 情報構造主導, 語順の自由度 (flexible word order) | Discourse- and topic-oriented packaging can override purely grammatical cues, shaping what is foregrounded, which reasoning path is taken, and what is implicitly treated as the objective or frame. | $z_{\text{process}} \rightsquigarrow$, $z_{\text{scope}} \downarrow$, $z_{\text{objective}}(\pm)$ |

**JP:** この資料に基づいて、
(1) 資料から直接言えること
(2) そこからの推測
を分けて、断定せずに説明してください。

**EN:** Based on this document, please separate (1) what is directly supported by the source and (2) what is inferred from it, without making definitive claims.

**Prompt D (Safety-prioritized framing).**

**JP:** 正確さよりも安全な理解を優先したいです。
現時点での見取り図と、未検証な点を分けて教えてください。

**EN:** I prioritize safe understanding over definitive correctness. Please separate the current overview from points that remain unverified.

Table B.1: Reference sheet: formal definitions of the eleven observer-side reporting axes in the Z-model.

| Dimension | Formal definition |
|---|---|
| $z_{\text{role}}$ | Adopted persona or social identity, regulating perspective and authority. |
| $z_{\text{task}}$ | Internal task schema inferred from prompt instructions. |
| $z_{\text{audience}}$ | Assumed reader expertise and informational needs. |
| $z_{\text{style}}$ | Stylistic register including politeness, formality, and affect. |
| $z_{\text{format}}$ | Expected output organization (lists, tables, code, etc.). |
| $z_{\text{length}}$ | Verbosity/brevity target influencing compression and expansion. |
| $z_{\text{process}}$ | Mode of reasoning (stepwise, exploratory, contrastive, etc.). |
| $z_{\text{scope}}$ | Breadth of conceptual coverage and alternative generation; marker breadth is not equated with perceived sufficiency. |
| $z_{\text{epistemic}}$ | Confidence calibration, hedging, and uncertainty expression. |
| $z_{\text{constraint}}$ | Explicit prohibitions delimiting allowable reasoning pathways. |
| $z_{\text{objective}}$ | Implicit optimization target (accuracy, creativity, critique, etc.). |

## B.2 Two-Condition Probing Protocol (A/B)

We operationalize the notion of a "response space" via observable output features: (i) degree of definitive closure, (ii) number of explicitly stated conditions, (iii) number of alternatives or competing explanations, and (iv) density of explicit uncertainty markers.

**Terminology note.** We use "2×2" only to describe the *presentation layout* that crosses (prompt condition vs. output features) under two probing conditions (A/B). This should not be interpreted as a factorial experimental design. Empirically, this is a paired comparison in which epistemic framing (A vs. B) is varied while other settings (model, sampling temperature, token limits) are held fixed.

## B.3 Sentence-Final Control as a Reliability Lever

In Japanese, epistemic stance is strongly grammaticalized at the sentence-final position through modality, politeness markers, and discourse particles. As a result, sentence-final expressions act as low-cost control signals that specify the intended interaction protocol.

Let $y$ denote a generated response, $c$ the propositional content of a prompt, and $s$ its epistemic stance. Model behavior can be abstracted as sampling from $p(y \mid c, s)$. Sentence-final expressions primarily modulate $s$ rather than $c$. Agreement-seeking formulations tend to sharpen the conditional distribution, concentrating probability mass on definitive outputs. Epistemic softeners broaden the effective support, increasing conditionality, alternatives, and explicit uncertainty.

## B.4 Agreement-Seeking and Premise-Fixing Expressions

Table B.2 lists representative Japanese expressions that often induce premature closure. These are not "stylistic mistakes"; rather, they function as interaction-level risk factors whose effects can be assessed with the output features above. The listed effects should be read as empirical tendencies and are context-dependent.

Table B.2: Agreement-seeking and premise-fixing expressions in Japanese and their typical effects on generative model outputs.

| No. | Japanese expression | Primary function | Why it induces premature closure | Typical effect on model outputs |
|---|---|---|---|---|
| 1 | 〜ですよね | Agreement seeking | Presupposes shared agreement, increasing the social cost of introducing exceptions | Model tends to comply and omit conditions or alternatives |
| 2 | 当然〜 | Premise fixation | Frames the proposition as beyond doubt, discouraging verification or counterexamples | More assertive, potentially overgeneralized responses |
| 3 | 〜に決まってますよね | Conclusion imposition | Shifts the task from exploration to justification of a fixed conclusion | Post-hoc rationalization with weak grounding becomes more likely |
| 4 | 〜で間違いないですよね | Refutation blocking | Makes counterexamples equivalent to contradiction, suppressing exploratory reasoning | High probability of binary "Yes" closure |
| 5 | 〜は常識ですよね | Normative authority | Penalizes uncertainty as incompetence, reducing epistemic humility | Either vague generalities or confident assertions |
| 6 | 言うまでもなく〜 | Explanation suppression | Encourages leaving assumptions implicit rather than articulated | Model fills gaps with inferred premises; hallucination risk may increase |
| 7 | つまり〜ですよね | Single-path reduction | Collapses multiple interpretations into one enforced summary | Alternative models and exceptions disappear |
| 8 | 要するに〜でしょ | Forced summarization | Fixes a conclusion mid-discourse, eliminating nuance | Oversimplified conclusions dominate |
| 9 | 結局〜なんですよね | Discourse termination | Signals that further elaboration is unwelcome | Conditional explanations are suppressed |
| 10 | 前提として〜 | Unexamined premise fixation | Treats the premise as given without validation | Reasoning proceeds on potentially fragile assumptions |
| 11 | 正しいですか？ | Binary framing | Reduces graded or conditional truth to a yes/no judgment | Encourages definitive, unqualified answers |
| 12 | 間違いですか？ | Adversarial framing | Casts negation as confrontation, discouraging nuance | Defensive or overly compliant responses |
| 13 | 結論だけ言って | Context stripping | Removes the opportunity to state assumptions, limits, or scope | Short but misleading answers become more likely |
| 14 | 一言で言うと？ | Excessive compression | Forces complex phenomena into a single proposition | Overgeneralization increases |
| 15 | 絶対に〜 | Uncertainty elimination | Demands certainty rarely justified in scientific contexts | Strong assertions coupled with elevated misinformation risk |
| 16 | 100%で答えて | Uncertainty prohibition | Explicitly disallows expressing uncertainty | Hallucination probability increases |
| 17 | 反論は不要 | Exploration suppression | Prohibits counterarguments and alternatives | One-sided and biased outputs |
| 18 | おかしくない？ | Affective confrontation | Shifts the task from content evaluation to attitude response | Defensive or appeasing behavior |
| 19 | 白黒つけて | Polarization framing | Disallows graded reasoning and scientific ambiguity | Opinionated response mode is encouraged |
| 20 | 常識的に考えて | Vague authority appeal | Introduces an unverifiable normative standard | Unsupported general claims become more likely |

## B.5 Premise-Fixing Mini-Probe (P2)

Motivated by Table B.2 (No. 1), we run a minimal A/B probe that appends "結論は正しいですよね？" to otherwise identical Japanese prompts. We treat this as an illustrative case study whose purpose is diagnostic: to check whether a premise-fixing cue produces measurable redistribution in our scope/stance proxies under the same black-box protocol and fixed decoding settings (Appendix B.2). This mini-probe is a legacy appendix illustration and is not part of the revised primary endpoint family; its shorthand labels (HEDGE, ALT, DEF, and PROP) are retained only to preserve the original diagnostic report.

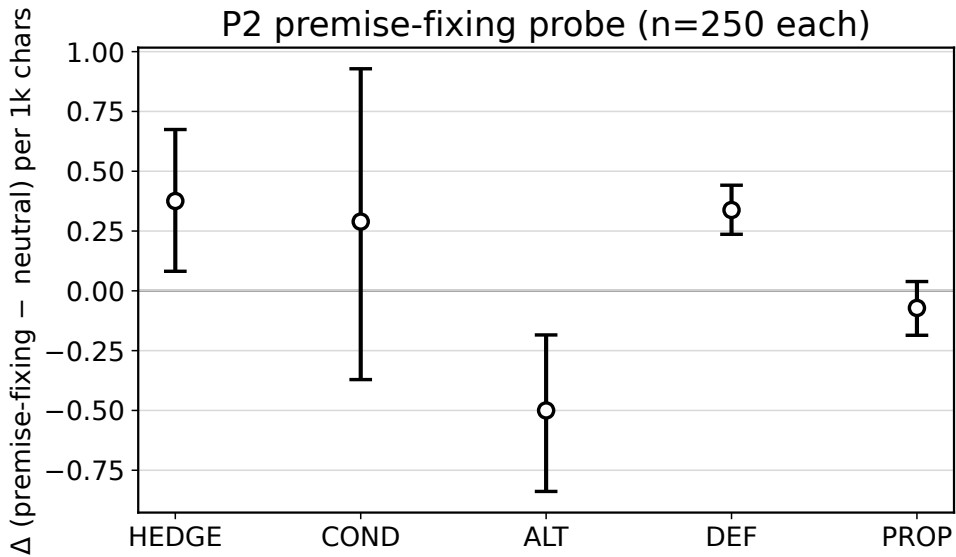

Figure B.1: Premise-fixing mini-probe (P2): point estimates and bootstrap 95% CIs for $\Delta = $ (premise-fixing − neutral) across five lexical proxies.

Figure B.1 visualizes $\Delta$ (premise-fixing minus neutral) with bootstrap 95% CIs for five lightweight lexical proxies (Appendix G), aggregated over $n = 250$ samples per condition. Table B.3 additionally reports Cohen's $d$ and histogram-based Jensen–Shannon divergence (JSD); because JSD is nonnegative, its bootstrap CI lower endpoint can be 0 (after rounding) for small shifts. As a lightweight semantic sanity check using a rubric aligned to these proxy definitions, we additionally run an LLM-as-a-Judge A/B direction test on the same paired outputs (Table B.4). Because the appended cue is intentionally minimal, per-proxy effect sizes are expected to be modest. Accordingly, while some bootstrap CIs include zero (e.g., COND), we focus on the structured redistribution signature across multiple proxies. The judge direction check provides a coarse semantic complement (most clearly consistent for HEDGE and COND) rather than a ground-truth calibration or a claim of statistical significance for any single coordinate. In this within-protocol probe, the premise-fixing cue yields a mixed redistribution signature: HEDGE increases and COND is directionally higher on average (more conditional framing) but its CI slightly overlaps zero, while ALT and PROP show only small net shifts. The DEF lexical proxy increases modestly, but the judge direction check does not show a corresponding increase in overall categorical tone, underscoring the limitations of surface indicators. Notably, HEDGE rises despite the agreement-seeking framing, suggesting that alignment and task semantics may counteract closure by introducing caveats. This mixed pattern is consistent with the Z-model view that premise-fixing can trigger coupled (and sometimes heterogeneous) redistribution effects rather than a single monotonic "confidence increase" across all indicators.

Table B.3: Premise-fixing mini-probe summary. $\Delta$ is computed as premise-fixing minus neutral; 95% CIs are bootstrap percentiles; JSD is histogram-based Jensen–Shannon divergence (Appendix B.2). Since JSD is nonnegative, the lower CI endpoint may be reported as 0 (after rounding), indicating bootstrap resamples with near-zero divergence.

| Metric | $\Delta$ (per 1k chars) | Cohen's $d$ | JSD |
|---|---|---|---|
| HEDGE | +0.339 [+0.070, +0.602] | +0.22 [+0.04, +0.39] | 0.023 [0.000, 0.046] |
| COND | +0.670 [-0.013, +1.317] | +0.18 [+0.00, +0.35] | 0.055 [0.018, 0.092] |
| ALT | -0.065 [-0.360, +0.232] | -0.04 [-0.21, +0.13] | 0.024 [0.000, 0.049] |
| DEF | +0.081 [+0.004, +0.164] | +0.18 [+0.01, +0.33] | 0.011 [0.000, 0.024] |
| PROP | -0.044 [-0.158, +0.070] | -0.07 [-0.26, +0.10] | 0.010 [0.000, 0.023] |

Table B.4: LLM-as-a-Judge A/B direction check for the premise-fixing mini-probe (P2), aligned to the proxy definitions in Appendix G. Judge: gpt-5-mini. A/B order randomized per pair ($n = 250$); ties allowed. Responses clipped (chars=800). Execution window: 2026-02-27 UTC.

| Property | Premise | Neutral | Tie | Premise win rate (excl. ties) | 95% CI |
|---|---|---|---|---|---|
| HEDGE | 117 | 64 | 69 | 0.646 | [0.574, 0.712] |
| DEF | 96 | 113 | 41 | 0.459 | [0.393, 0.527] |
| COND | 123 | 78 | 49 | 0.612 | [0.543, 0.677] |
| ALT | 88 | 93 | 69 | 0.486 | [0.414, 0.559] |
| PROP | 91 | 77 | 82 | 0.542 | [0.466, 0.615] |

## C  Minimal Sanity Check of the Z-model

This appendix reports a minimal sanity check designed to assess whether selected reporting axes can be linked to observable properties of model outputs, rather than serving solely as a post-hoc conceptual categorization.

The goal of this experiment is explicitly diagnostic rather than evaluative: we do not aim to establish statistical significance, benchmark performance, or estimate effect sizes. Instead, we examine whether directional expectations along selected reporting axes are reflected in simple, automatically measurable output indicators under controlled prompt perturbations.

Note that all reported $\Delta$ values correspond to changes in observable proxies. For epistemic indicators, increased hedging frequency reflects *lower epistemic commitment* (i.e., $z_{\text{epistemic}} \downarrow$), whereas strong assertions correspond to higher epistemic commitment ($z_{\text{epistemic}} \uparrow$).

### C.1  Protocol

We fixed a base prompt in Japanese that naturally elicits epistemic caution, moderate scope, and structured reasoning. From this base prompt, we generated a small set of probe prompts by appending short, commonly used prompt fragments (e.g., "箇条書きで" [in bullet points], "断定せずに" [without making definitive claims]).

Each fragment was chosen to primarily target a single Z-dimension (e.g., `z_format`, `z_process`, `z_epistemic`, or `z_scope`), while acknowledging that secondary effects may occur. To reduce confounding interactions, we evaluated only the specified directional expectation for the primary target axis of each fragment.

For each prompt variant, we generated model outputs under two decoding temperatures ($T = 0.2$ and $T = 0.8$). All other generation parameters were held constant. No prompt tuning or example-based conditioning was used.

### C.2 Metrics

We employed lightweight, automatically measurable output proxies corresponding to the targeted reporting axes:

- **Process indicators**: presence of explicit step markers, enumerations, contrastive segmentation, or other structured reasoning cues.
- **Format indicators**: presence of explicit list-item markers (e.g., bullets or numbering) as a proxy for list-style formatting.
- **Epistemic indicators**: frequency of hedging expressions, uncertainty markers, or modal qualifiers.
- **Scope indicators**: occurrence of alternative viewpoints, conditions/exceptions, or explicit enumeration of multiple cases.

For each probe prompt, we assessed whether the output exhibited an increase (+), decrease (−), or no clear change (0) in the relevant indicator relative to the base prompt. Only the sign of the directional change was recorded; absolute magnitudes were not analyzed.

### C.3 Findings

Table C.1 summarizes the observed directional alignment between specified reporting axes and measured output indicators.

Across decoding temperatures, several fragments show consistent directional effects, most notably for process-related and epistemic dimensions. Scope-related effects appear more sensitive to both prompt context and temperature, but nonetheless exhibit interpretable trends in the specified direction.

These observations are consistent with the narrower claim that selected reporting axes can be operationally linked to observable output indicators. Crucially, this sanity check is not presented as empirical validation of the Z-model. Rather, it establishes observational grounding and falsifiability: the vocabulary yields concrete directional expectations that can, in principle, be tested and potentially refuted using simple output indicators.

**Why Japanese prompts?** We focus on Japanese as one diagnostic setting because epistemic stance, politeness, and agreement can be grammatically encoded at the sentence-final position. This motivates lightweight probing, but does not imply that the effects are unique to Japanese or stronger than in all other languages.

**Why these proxies?** The selected proxies are intentionally lightweight and surface-level, as the goal is not to infer latent states directly, but to test whether the Z-model vocabulary yields falsifiable directional expectations that manifest in simple, automatically detectable output properties.

Table C.1: Minimal sanity check: directional alignment between specified observer-side reporting axes and observable output indicators. For each prompt fragment, only the primary target reporting axis is evaluated. Symbols indicate the direction of change in the chosen observable proxy relative to the base prompt. **Importantly, "+" and "−" denote changes in the observable proxy, not in the observer-side $z$-axis itself. For $z_{\text{epistemic}}$, the proxy is hedging frequency, so "+" corresponds to increased hedging and thus lower epistemic commitment ($z_{\text{epistemic}} \downarrow$). For $z_{\text{format}}$, the proxy is the presence/count of explicit list-item markers, so "+" corresponds to more list-style formatting.**

| Prompt fragment | Target reporting axis | $\Delta$ ($T = 0.2$) | $\Delta$ ($T = 0.8$) |
|---|---|---|---|
| 箇条書きで | $z_{\text{format}}$ | + | + |
| ステップごとに | $z_{\text{process}}$ | + | + |
| 断定せずに | $z_{\text{epistemic}}$ | + | + |
| 慎重に | $z_{\text{epistemic}}$ | + | 0 |
| Ａ と Ｂ を比較して | $z_{\text{process}}$ | + | + |
| 代表例を挙げて | $z_{\text{scope}}$ | + | 0 |

# D    Detailed Numerical Tables for the Main Politeness Probe

This appendix collects numerical tables that were moved out of the main text to reduce table density. The main text uses forest plots and compact narrative summaries; the exact deltas and exploratory endpoints remain available here for audit and protocol re-execution.

Table D.1: Primary politeness-probe endpoints with topic-aware hierarchical bootstrap confidence intervals. Deltas are polite−plain, rates are per 1,000 characters. $q$ is Benjamini–Hochberg FDR over the eight revision-specified primary tests (four metrics × two languages).

| Primary endpoint | $\Delta_{\text{JP}}$ | $q_{\text{JP}}$ | $\Delta_{\text{EN}}$ | $q_{\text{EN}}$ |
|---|---|---|---|---|
| HEDGE_EPI | -0.004 [-0.087, 0.062] | 1.000 | 0.237 [0.078, 0.417] | 0.003 |
| HEDGE_GEN | -0.448 [-0.925, 0.039] | 0.140 | -0.003 [-0.202, 0.182] | 1.000 |
| COND | 0.826 [0.287, 1.343] | 0.016 | -0.011 [-0.295, 0.270] | 1.000 |
| ALT_CONTR | -0.163 [-0.288, -0.018] | 0.079 | 0.000 [0.000, 0.000] | 1.000 |

*Note.* EN ALT_CONTR has zero or near-zero variation under this specific English politeness protocol. It is retained because the endpoint family was fixed symmetrically across languages in the revision; the zero entries should be read as transparent nonactivation of the contrastive-alternative dictionary, not as evidence for a calibrated semantic absence of alternative reasoning.

Table D.2: Topic-aware bootstrap Cohen's $d$ for the primary endpoints. Deltas are reported in Table D.1.

| Primary endpoint | $d_{\text{JP}}$ | $d_{\text{EN}}$ |
|---|---|---|
| HEDGE_EPI | -0.010 [-0.188, 0.197] | 0.503 [0.203, 0.875] |
| HEDGE_GEN | -0.216 [-0.525, 0.018] | -0.003 [-0.230, 0.234] |
| COND | 0.289 [0.103, 0.532] | -0.009 [-0.283, 0.258] |
| ALT_CONTR | -0.288 [-0.609, -0.030] | 0.000 [0.000, 0.000] |

Table D.3: Raw-count robustness for the primary politeness-probe endpoints. Deltas are polite−plain raw marker-count differences per output, with topic-aware hierarchical bootstrap 95% CIs. $q$ is Benjamini–Hochberg FDR over the eight raw-count primary checks.

| Endpoint | $\Delta$ raw count | $q$ |
|---|---|---|
| JP HEDGE_EPI | 0.016 [-0.024, 0.072] | 0.963 |
| JP HEDGE_GEN | 0.024 [-0.304, 0.404] | 1.000 |
| JP COND | 0.932 [0.508, 1.372] | 0.010 |
| JP ALT_CONTR | -0.032 [-0.136, 0.096] | 0.963 |
| EN HEDGE_EPI | 0.236 [0.076, 0.428] | 0.011 |
| EN HEDGE_GEN | 0.236 [-0.084, 0.580] | 0.442 |
| EN COND | -0.056 [-0.384, 0.272] | 0.963 |
| EN ALT_CONTR | 0.000 [0.000, 0.000] | 1.000 |

Table D.4: Exploratory and structural endpoints for the politeness probe. Deltas are polite−plain with topic-aware hierarchical bootstrap 95% CIs. These metrics are reported for interpretation and robustness, not as primary evidence.

| Exploratory endpoint | $\Delta_{\mathrm{JP}}$ | $\Delta_{\mathrm{EN}}$ |
|---|---|---|
| ADDITIVE | 0.273 [-0.082, 0.690] | 0.021 [-0.210, 0.282] |
| DEFINITE | 0.066 [-0.019, 0.159] | -0.003 [-0.021, 0.016] |
| PROPENSITY | -0.163 [-0.561, 0.278] | 0.148 [-0.075, 0.445] |
| CHARS | 43.496 [30.491, 56.260] | -24.652 [-40.856, -4.039] |
| SENTENCES | 2.460 [1.660, 3.244] | 0.764 [0.280, 1.304] |
| LIST_ITEMS | -1.372 [-2.784, -0.332] | -0.732 [-1.276, -0.404] |

# E   Language-Qualified Reader-Check Diagnostics

This appendix reports diagnostics for the language-qualified direct-reader check summarized in the main politeness-probe results. The analysis separates direct readings from possible translation-assisted English judgments. This distinction matters because machine translation can alter precisely the cues under study, including hedging, conditionality, and discourse-scope markers. Reader-set membership is determined by direct-reading eligibility for the corresponding language, not by the direction or magnitude of the ratings. In the artifact bundle, individual reader columns are anonymized as R1–R4; a role-only eligibility key records which anonymous readers are included in the direct-Japanese and direct-English checks without exposing names, initials, email addresses, or organizational identifiers.

**Table E.1: Reader-set eligibility and use.** Before considering the later R5 form, the reader-set rule was fixed by language eligibility: the main analysis uses only judgments from readers who directly read the corresponding language output, and translation-assisted English ratings are treated as sensitivity material only. One additional completed form, denoted R5-unverified below, is excluded from all analyses because direct-reading eligibility could not be verified; the artifact bundle retains only a note documenting this exclusion and does not include the raw unverified form.

| Reader set | Language output | Readers | Use in analysis |
|---|---|---|---|
| Direct Japanese | JP | R1, R2, R3, R4 | Main JP reader check ($n = 4$) |
| Direct English | EN | R1, R2 | Main EN reader check ($n = 2$) |
| Translation-assisted English | EN | R3, R4 | Appendix sensitivity only |
| R5-unverified additional form | JP/EN | R5 | Excluded; note-only in bundle |

**Table E.2: Direct-reader deltas with multiple-comparison adjustment.** Deltas are polite−plain on the 1–5 reader scale. The $q$ column is Benjamini–Hochberg adjusted over the eight direct-reader comparisons, but is reported

only for auditability and not as inferential support. Because the reader study is small and inter-reader agreement is low, effect directions and intervals are treated as exploratory context rather than confirmatory semantic validation.

| Endpoint | $\Delta$ reader rating | $q$ |
|---|---|---|
| JP Q1 | -0.02 [-0.17, 0.12] | 0.84 |
| JP Q2 | 0.04 [-0.14, 0.24] | 0.84 |
| JP Q3 | -0.06 [-0.19, 0.08] | 0.58 |
| JP Q4 | 0.19 [-0.04, 0.41] | 0.23 |
| EN Q1 | 0.09 [-0.22, 0.40] | 0.79 |
| EN Q2 | 0.29 [0.09, 0.48] | 0.04 |
| EN Q3 | 0.33 [0.05, 0.60] | 0.10 |
| EN Q4 | 0.26 [-0.01, 0.52] | 0.16 |

**Table E.3: Agreement diagnostics by direct-reader subset.** Exact agreement is strict five-point agreement; "within 1" counts scores differing by at most one point. QWK denotes quadratic weighted kappa. The within-one agreement rates indicate that readers often used nearby scale regions, but the QWK values show that the ratings are not a calibrated semantic measurement instrument. This is why the reader study is used only as coarse exploratory context.

| Question | JP within 1 / QWK | JP exact | EN within 1 / QWK | EN exact |
|---|---|---|---|---|
| Q1 categorical tone | 0.86 / -0.01 | 0.28 | 0.78 / -0.06 | 0.33 |
| Q2 uncertainty/qualification | 0.87 / 0.04 | 0.37 | 0.68 / 0.00 | 0.24 |
| Q3 scope/sufficiency | 0.91 / 0.04 | 0.40 | 0.86 / 0.05 | 0.40 |
| Q4 explanatory scaffolding | 0.60 / -0.02 | 0.21 | 0.37 / -0.01 | 0.12 |

**Table E.4: Translation-assisted English sensitivity.** When the two additional English judgments are pooled with the direct-English readers, the English reader effects attenuate. Because these additional judgments may have passed through a machine-translation layer, they are not used as the primary English reader check. This sensitivity analysis is included to show how the direct-reader criterion affects the descriptive reader results.

| Human endpoint | All-four EN $\Delta$ polite–plain | 95% CI |
|---|---|---|
| Q1 categorical tone | -0.04 | [-0.38, 0.28] |
| Q2 uncertainty/qualification | 0.13 | [-0.01, 0.27] |
| Q3 scope/sufficiency | 0.16 | [0.01, 0.31] |
| Q4 explanatory scaffolding | 0.21 | [-0.03, 0.45] |

**Table E.5: Measurement interpretation for scope-related checks.** The reader study documents where marker quantity and perceived sufficiency may diverge. This distinction is conceptual and measurement-theoretic; it is not inferred from low reader correlations.

| Measurement layer | Operational signal | Interpretation in this paper |
|---|---|---|
| Lexical proxy | COND, ALT_CONTR, ADDITIVE | Enumerative breadth / marker redistribution |
| Structural proxy | sentences, list items, characters | Explanation quantity / realization form |
| Reader item Q3 | scope/sufficiency rating | Relevance-weighted perceived coverage |
| Reader item Q4 | scaffolding rating | Perceived explanatory support |

**Table E.6: Condition-stratified proxy–reader Spearman correlations.** For transparency, we report selected correlations between automatic surface proxies and direct-reader ratings within each language and condition. These values are descriptive only. They are not used as convergent-validity evidence, and low, null, or negative correlations are not interpreted as proof of construct separation, because the reader instrument itself has low inter-reader agreement. Dashes indicate undefined Spearman correlations due to zero variance or insufficient variation in one of the two variables within that subset.

| Lang. | Cond. | Proxy → reader item | $\rho$ | $n$ |
|-------|-------|---------------------|--------|-----|
| JP | plain | DEFINITE → Q1 definitiveness | -0.09 | 50 |
| JP | plain | HEDGE_EPI → Q2 uncertainty | -0.37 | 50 |
| JP | plain | HEDGE_GEN → Q2 uncertainty | -0.15 | 50 |
| JP | plain | COND → Q3 scope/suff. | -0.05 | 50 |
| JP | plain | ALT_CONTR → Q3 scope/suff. | +0.23 | 50 |
| JP | plain | ADDITIVE → Q3 scope/suff. | +0.04 | 50 |
| JP | plain | LIST_ITEMS → Q4 scaffolding | +0.05 | 50 |
| JP | plain | SENTENCES → Q4 scaffolding | +0.02 | 50 |
| JP | polite | DEFINITE → Q1 definitiveness | +0.06 | 50 |
| JP | polite | HEDGE_EPI → Q2 uncertainty | -0.05 | 50 |
| JP | polite | HEDGE_GEN → Q2 uncertainty | -0.27 | 50 |
| JP | polite | COND → Q3 scope/suff. | +0.26 | 50 |
| JP | polite | ALT_CONTR → Q3 scope/suff. | -0.01 | 50 |
| JP | polite | ADDITIVE → Q3 scope/suff. | -0.12 | 50 |
| JP | polite | LIST_ITEMS → Q4 scaffolding | +0.05 | 50 |
| JP | polite | SENTENCES → Q4 scaffolding | -0.06 | 50 |
| EN | plain | DEFINITE → Q1 definitiveness | -0.08 | 50 |
| EN | plain | HEDGE_EPI → Q2 uncertainty | +0.10 | 50 |
| EN | plain | HEDGE_GEN → Q2 uncertainty | +0.07 | 50 |
| EN | plain | COND → Q3 scope/suff. | -0.00 | 50 |
| EN | plain | ALT_CONTR → Q3 scope/suff. | – | 50 |
| EN | plain | ADDITIVE → Q3 scope/suff. | +0.05 | 50 |
| EN | plain | LIST_ITEMS → Q4 scaffolding | +0.14 | 50 |
| EN | plain | SENTENCES → Q4 scaffolding | +0.19 | 50 |
| EN | polite | DEFINITE → Q1 definitiveness | – | 50 |
| EN | polite | HEDGE_EPI → Q2 uncertainty | +0.01 | 50 |
| EN | polite | HEDGE_GEN → Q2 uncertainty | +0.02 | 50 |
| EN | polite | COND → Q3 scope/suff. | +0.37 | 50 |
| EN | polite | ALT_CONTR → Q3 scope/suff. | – | 50 |
| EN | polite | ADDITIVE → Q3 scope/suff. | -0.56 | 50 |
| EN | polite | LIST_ITEMS → Q4 scaffolding | -0.16 | 50 |
| EN | polite | SENTENCES → Q4 scaffolding | +0.22 | 50 |

Dashes indicate undefined Spearman correlations because one variable has zero variance or too little variation within that language–condition subset. Correlations are descriptive and are not used as convergent-validity evidence.

**Table E.7: Rubric-based LLM-judge proxy correlation check.** As a separate rubric-based diagnostic, we scored the 1000 main politeness-probe outputs with an LLM judge and computed condition-stratified Spearman correlations between automatic surface proxies and judge ratings. The judge model was `gpt-5-mini-2025-08-07`; the Batch API run completed on 2026-06-06 UTC (23:45–23:55), and the request/response JSONL files are included in the artifact bundle. This table is included to make the requested proxy–semantic comparison explicit, not as confirmatory validation and not as a substitute for human annotation. The results are mixed: EN HEDGE_GEN correlates moderately to strongly with judge-rated uncertainty/qualification, whereas HEDGE_EPI is condition-dependent; JP COND and ALT_CONTR show small positive associations with judge-rated scope/sufficiency; several other pairings are weak, negative, or undefined because of sparse proxy variation. The judge ratings also show saturation for some scope/scaffolding items, so this check is best read as a rubric-based descriptive diagnostic rather than a replacement for human perception. We therefore retain the interpretation that lexical indicators are transparent surface-marker proxies rather than calibrated semantic measurements.

| Lang. | Cond. | Proxy → judge item | $\rho$ | $n$ |
|---|---|---|---|---|
| JP | plain | DEFINITE → Q1_definitiveness | +0.00 | 250 |
| JP | plain | HEDGE_EPI → Q2_uncertainty_qualification | +0.02 | 250 |
| JP | plain | HEDGE_GEN → Q2_uncertainty_qualification | +0.02 | 250 |
| JP | plain | COND → Q3_scope_sufficiency | +0.19 | 250 |
| JP | plain | ALT_CONTR → Q3_scope_sufficiency | +0.20 | 250 |
| JP | plain | ADDITIVE → Q3_scope_sufficiency | +0.06 | 250 |
| JP | plain | LIST_ITEMS → Q4_explanatory_scaffolding | -0.17 | 250 |
| JP | plain | SENTENCES → Q4_explanatory_scaffolding | +0.04 | 250 |
| JP | polite | DEFINITE → Q1_definitiveness | -0.11 | 250 |
| JP | polite | HEDGE_EPI → Q2_uncertainty_qualification | +0.04 | 250 |
| JP | polite | HEDGE_GEN → Q2_uncertainty_qualification | -0.06 | 250 |
| JP | polite | COND → Q3_scope_sufficiency | +0.19 | 250 |
| JP | polite | ALT_CONTR → Q3_scope_sufficiency | +0.20 | 250 |
| JP | polite | ADDITIVE → Q3_scope_sufficiency | +0.06 | 250 |
| JP | polite | LIST_ITEMS → Q4_explanatory_scaffolding | +0.11 | 250 |
| JP | polite | SENTENCES → Q4_explanatory_scaffolding | +0.03 | 250 |
| EN | plain | DEFINITE → Q1_definitiveness | -0.06 | 250 |
| EN | plain | HEDGE_EPI → Q2_uncertainty_qualification | -0.07 | 250 |
| EN | plain | HEDGE_GEN → Q2_uncertainty_qualification | +0.52 | 250 |
| EN | plain | COND → Q3_scope_sufficiency | +0.03 | 250 |
| EN | plain | ALT_CONTR → Q3_scope_sufficiency | – | 250 |
| EN | plain | ADDITIVE → Q3_scope_sufficiency | -0.01 | 250 |
| EN | plain | LIST_ITEMS → Q4_explanatory_scaffolding | +0.13 | 250 |
| EN | plain | SENTENCES → Q4_explanatory_scaffolding | +0.38 | 250 |
| EN | polite | DEFINITE → Q1_definitiveness | +0.13 | 250 |
| EN | polite | HEDGE_EPI → Q2_uncertainty_qualification | +0.19 | 250 |
| EN | polite | HEDGE_GEN → Q2_uncertainty_qualification | +0.56 | 250 |
| EN | polite | COND → Q3_scope_sufficiency | -0.19 | 250 |
| EN | polite | ALT_CONTR → Q3_scope_sufficiency | – | 250 |
| EN | polite | ADDITIVE → Q3_scope_sufficiency | +0.12 | 250 |
| EN | polite | LIST_ITEMS → Q4_explanatory_scaffolding | -0.07 | 250 |
| EN | polite | SENTENCES → Q4_explanatory_scaffolding | +0.07 | 250 |

Dashes indicate undefined correlations due to zero or insufficient variation. LLM-judge correlations are rubric-based diagnostics and are not used as substitutes for human annotation.

**Additional completed form with uncertain eligibility.** One additional completed reader form (R5-unverified) was received after the main reader-analysis rule had been fixed. Because direct-reading eligibility for both languages could not be verified, this form is excluded from all reader analyses. The exclusion is based solely on language eligibility, not on the direction or magnitude of the ratings. For privacy and claim–artifact consistency, the artifact bundle

retains only an exclusion note and does not include the raw unverified form; no table, figure, or conclusion in the paper depends on it.

## F   Additional Framing→Reasoning Probe (Exp. C)

This appendix gives the implementation details for the supplementary audience-framing probe introduced in Section 3.5. The goal is to add one cross-group pathway beyond the main Expression→Epistemic/Scope politeness probe, while preserving the same black-box, distribution-level measurement philosophy.

**Prompt design.**   The primary manipulation is $z_{\text{audience}}$. For each topic and language, the neutral condition asks for a direct explanation of the concept, whereas the novice-audience condition asks for an explanation for a non-specialist or beginner reader. All other constraints are fixed. The Japanese audience cue is「初学者にもわかるように」(literal gloss: "so that even a beginner can understand"); the English cue is "for a non-specialist reader."

> **EN neutral:** Explain the following concept: `{topic}`. Constraints: answer in exactly one paragraph; do not use bullet points, numbered lists, tables, or headings; write about 6–8 sentences; use a neutral explanatory tone; avoid formulas unless necessary; do not mention these constraints.
> **EN novice-audience:** Explain the following concept for a non-specialist reader: `{topic}`. [same constraints]
> **JP neutral:** 次の概念を説明してください：`{topic}`。制約：1つの段落だけで答え、箇条書き、番号付きリスト、表、見出しは使わないでください。6〜8文程度にしてください。中立的な説明調にしてください。必要がなければ数式は使わないでください。これらの制約には言及しないでください。
> **JP novice-audience:** 次の概念を初学者にもわかるように説明してください：`{topic}`。[same constraints]

**Topics and model settings.**   We reuse five benign technical topics: noise/SNR/averaging, overfitting and validation data, caching and latency, correlation versus causation, and gradient descent/learning rate. The API-served model is the pinned OpenAI API snapshot `gpt-5.2-2025-12-11` (reported alias: GPT-5.2), accessed during 2026-05-23–24 UTC, with temperature $= 0.7$, top_p $= 1.0$, max_tokens $= 512$, and $n = 50$ generations per topic, language, and condition. The resulting design contains $5 \times 2 \times 2 \times 50 = 1000$ generations. All 1000 planned calls completed without API errors in the logged run.

**Reasoning-presentation proxy dictionary.**   For Exp. C, we add a small set of reasoning-presentation indicators to the core A/B dictionary in Appendix G:

```
definition: means, refers to, is called, とは, というのは
example: for example, e.g., for instance, 例えば, たとえば, 例として, 具体例
analogy: like, similar to, as if, imagine, analogy, analogous, たとえると, 例えると, ような, よ
うに, 似て, イメージ, 比喩
causal: because, therefore, so, as a result, なぜなら, そのため, したがって, 結果として
contrast: however, whereas, in contrast, 一方, ただし, しかし
sequence: first, then, finally, まず, 次に, 最後に
```

The aggregate `reasoning_scaffold` proxy is the sum of definition, example, analogy, causal, contrast, and sequence counts. As in the main experiments, lexical quantities are normalized per 1,000 Unicode characters after NFKC normalization, while raw counts are also checked to guard against length-normalization artifacts.

Table F.1: Exp. C raw-count deltas recomputed from raw audience-framing generations. Deltas are novice/non-specialist audience minus neutral; topic-aware bootstrap 95% CIs are shown in brackets.

| Metric | $\Delta_{\text{EN}}$ | $\Delta_{\text{JP}}$ |
|---|---|---|
| Reasoning scaffold | +0.460 [+0.084, +0.828] | +1.024 [+0.612, +1.416] |
| Definition | +0.164 [+0.020, +0.344] | +0.168 [−0.060, +0.460] |
| Analogy | +0.588 [+0.288, +0.824] | +0.484 [+0.208, +0.764] |
| Conditional | +0.340 [−0.116, +0.868] | −0.440 [−1.008, +0.056] |
| chars | −56.640 [−95.624, −18.968] | −18.432 [−28.184, −8.632] |
| sentences | −0.024 [−0.064, +0.012] | −0.004 [−0.024, +0.008] |
| list_items | 0.000 [0.000, 0.000] | 0.000 [0.000, 0.000] |

Table F.2: Exp. C stricter analogy-dictionary check recomputed from raw audience-framing generations. The stricter dictionary excludes broad analogy markers from the original analogy proxy. Deltas are novice/non-specialist audience minus neutral.

| Metric | $\Delta_{\text{EN}}$ | $\Delta_{\text{JP}}$ |
|---|---|---|
| Strict analogy count | +0.204 [+0.012, +0.488] | +0.340 [+0.040, +0.656] |
| Strict analogy / 1k chars | +0.183 [+0.010, +0.434] | +0.818 [+0.085, +1.593] |

**Raw-count check.** Table F.1 reports raw-count deltas recomputed from the raw Exp. C generations. The raw-derived summaries show positive audience-framing movement in analogy-oriented and explanatory-scaffolding markers in both languages. The effect is not a uniform increase in every reasoning-related proxy: definition markers are stronger in English than Japanese, and conditional markers are not consistently positive. This is why the main text frames Exp. C as localized reasoning-presentation redistribution rather than broad validation of the Framing or Reasoning macro-groups. The sign of COND is cue-dependent: Japanese politeness in Exp. A increases conditional framing, whereas novice-audience framing in Exp. C shifts more toward analogy and scaffolding than conditionality. We therefore treat COND as a surface discourse marker, not as a monotonic measure of a single semantic construct.

**Stricter analogy dictionary.** The broad analogy dictionary intentionally captures audience-oriented explanatory devices, but some entries are broad (especially English "like" and Japanese「ように／ような」). As a robustness check, we recompute analogy deltas after removing these broad markers and retaining the remaining analogy cues. The effect remains positive in both languages (Table F.2), supporting the interpretation that the audience cue increases analogy-oriented explanation rather than merely exploiting one broad lexical item.

## G  Minimal Feature Dictionary for A/B Probing (JP/EN)

**PROPENSITY** captures exploratory or forward-looking markers (e.g., "suggests," "may lead to") and serves as an auxiliary proxy for response openness.

**Purpose.** We provide a lightweight, reproducible feature dictionary to operationalize the output features used in the A/B probing protocol (Appendix B.2): (i) definitive closure, (ii) # conditions, (iii) contrastive alternatives versus additive markers, and (iv) explicit uncertainty markers.

**Counting rule (implementation-agnostic).** Given an output text $y$, we compute each feature as the number of non-overlapping matches of the corresponding regex pattern. For English patterns, we use case-insensitive matching. For list items, we use multiline mode. We recommend Unicode normalization (e.g., NFKC) prior to matching to reduce full-width / half-width variance.

**Normalization and bootstrap procedure.** For each output $y$, let $\ell(y)$ be the number of Unicode characters (after normalization) and let $c_k(y)$ be the match count for feature $k$. For lexical features, we compute a per-output rate $r_k(y) = 1000\, c_k(y)/\ell(y)$ and report mean rates aggregated over outputs. For the main politeness probe, per-condition differences (polite$-$plain) are estimated with a topic-aware hierarchical bootstrap: sample the five topics with replacement, and within each selected topic resample generations with replacement inside each condition before recomputing the difference. This preserves the topic–condition nesting that is ignored by an output-level iid bootstrap. We report 95% percentile intervals and Benjamini–Hochberg FDR-adjusted $q$ values for the revision-specified primary endpoints. For structural quantities (CHARS, SENTENCES, LIST_ITEMS), we report mean per-output differences without length normalization and treat them as exploratory.

### G.1 Core lexical indicators (JP/EN)

The revision-specified primary lexical endpoints are HEDGE_EPI, HEDGE_GEN, COND, and ALT_CONTR. ALT_CONTR contains contrastive or alternative-reasoning markers only. Simple enumerative markers are separated into ADDITIVE and reported as exploratory. We report the earlier ASSERT proxy under the clearer name DEFINITE and treat it as exploratory because lexical definitiveness is a coarse proxy for categorical tone. We additionally report PROPENSITY as an auxiliary indicator of response openness.

**Note on interpretation.** All lexical measures are surface-marker counts. They are intended for within-protocol directional comparison and are complemented by the language-qualified direct-reader check; neither source is treated as a calibrated semantic measurement.

**Raw/derived ALT provenance.** The raw main-probe JSONL retains legacy combined `alt_markers`/`alt_per_1k_chars` fields from an earlier broad alternative/additive dictionary for backward compatibility. The revised manuscript does not use those legacy raw ALT fields as primary endpoints. The canonical revision-level feature table is `data/main_politeness /derived/response_level_features.csv`, where ALT_CONTR and ADDITIVE are recomputed separately from the Appendix G dictionaries.

Table G.1: Representative lexical cue lists for A/B probing (JP/EN). The table summarizes human-readable lexical cues used to define lightweight *lexical observable proxies*. Formal regex definitions used for implementation are provided separately in Appendix G.2.

| Feature | Japanese cue list (JP) | English cue list (EN) |
|---|---|---|
| HEDGE_EPI | かもしれない/ません，可能性がある/あります，おそらく，多分/たぶん，不明，未検証，断定できない/ません，わからない/ません，推測，と思われる/ます，と考えられる/ます | `may, might, could, possibly, perhaps, uncertain, unknown, not sure, not clear, it is/it's possible, cannot conclude, can't conclude` |
| HEDGE_GEN | 一般に/的に，典型的に，多くの/は，よくある，傾向がある/にある，通常は，大抵 | `typically, in general, generally speaking, usually, often, tends to, in many cases` |
| COND | 場合，なら，とき/時，ただし/但し，例外，条件，前提，依存，次第で，によって，により，限り | `if, when, unless, provided that, assuming, in case, depends, depending on, except, exception, under the condition` |
| ALT_CONTR | 一方で，他方，別の可能性，別の見方，別の観点，代替案，異なる説明，選択肢 | `alternatively, on the other hand, an alternative, alternative view/explanation, another possibility/view/perspective/option, in contrast, by contrast` |
| ADDITIVE | また，さらに，加えて，他にも，もう一つ，補足すると | `also, additionally, furthermore, moreover, in addition` |
| DEFINITE | 必ず，絶対，間違いない，確実，当然，に決まっている/る/ます，に違いない，100\%，断言 | `definitely, certainly, must, always, undoubtedly, no doubt, 100\%` |
| PROPENSITY | 可能性，示唆，考えられ，想定され，起こりうる，つながる/ながる，導く，寄与，促す，見込まれ，うる，得る，検討でき | `suggests, may/could/might lead to, can be considered, is consistent with, indicates, implies, may imply, can help, may help` |

## G.2 Formal regex definitions for lexical and structural proxies

The following verbatim blocks provide the exact regular expressions used in our implementation (Appendix G.1 gives readable cue lists).

```
HEDGE_EPI_JP = (?: かもしれ (?: ない|ません)|可能性 (?: が|も)?(?:ある|あります)?|おそらく|多分|たぶん|不明|未検証|
        断定でき (?: ない|ません)|わから (?: ない|ません)|推測|と思われ (?: る|ます)|と考えられ (?: る|ます))
HEDGE_EPI_EN = (?i)(?:\bmay\b|\bmight\b|\bcould\b|\bpossibly\b|\bperhaps\b|\buncertain\b|\bunknown\b|
        \bnot\s+sure\b|\bnot\s+clear\b|\bit\s+(?:is|'s)\s+possible\b|\bcannot\s+conclude\b|\bcan't\s+conclude\b)

HEDGE_GEN_JP = (?: 一般 (?: に|的に)|典型的 (?: に)?|多く (?: の|は)|よく (?: ある)?|傾向 (?: が|に)?(?:ある)?|通常 (?: は)?|大抵)
HEDGE_GEN_EN =
(?i)(?:\btypically\b|\bin\s+general\b|\bgenerally\s+speaking\b|\busually\b|\boften\b|\btends?\s+to\b|\bin\s+many\s+cases\b)

COND_JP = (?: 場合|なら|とき|時|ただし|但し|例外|条件|前提|依存|次第で|によって|により|限り)
COND_EN = (?i)(?:\bif\b|\bwhen\b|\bunless\b|\bprovided\s+that\b|\bassuming\b|\bin\s+case\b|\bdepends?\b|\bdepending\s+on\b|\ ⌋
bexcept\b|\bexception\b|\bunder\s+the\s+condition\b)

ALT_CONTR_JP = (?:一方で|他方|別(?:の)?(?:可能性|見方|観点|選択肢|方法|説明)|代替(?:案|策)|異なる(?:見方|観点|説明)|観点(?:から|では
))
ALT_CONTR_EN = (?i)(?:\balternatively\b|\ban\s+alternative\b|\balternative\s+(?:view|explanation|hypothesis|option|approach) ⌋
\b|\banother\s+(?:possibility|view|perspective|option|explanation|hypothesis|approach|way)\b|\bon\s+the\s+other\s+hand\b|\bi ⌋
n\s+contrast\b|\bby\s+contrast\b|\bfrom\s+another\s+perspective\b)
ADDITIVE_JP = (?: また|さらに|加えて|他にも|もう一つ|補足すると|加えると)
ADDITIVE_EN = (?i)(?:\balso\b|\badditionally\b|\bfurthermore\b|\bmoreover\b|\bin\s+addition\b)

DEFINITE_JP = (?: 必ず|絶対|間違いない|確実|当然|に決まって (?: い (?: る|ます)|る)?|に違いない|100\%|断言)
DEFINITE_EN = (?i)(?:\bdefinitely\b|\bcertainly\b|\bmust\b|\balways\b|\bundoubtedly\b|\bno\s+doubt\b|\b100\%\b)
```

```
# Legacy aliases used in earlier scripts:
ASSERT_JP = DEFINITE_JP
ASSERT_EN = DEFINITE_EN

PROPENSITY_JP = (?: 可能性|示唆|考えられ|想定され|起こりうる|つながる|ながる|導く|寄与|促す|
        見込まれ|〜?うる|〜?得る|検討でき)
PROPENSITY_EN = (?i)(?:\bsuggests?\b|\bmay\s+lead\s+to\b|\bcould\s+lead\s+to\b|\bmight\s+lead\s+to\b|
        \bcan\s+be\s+considered\b|\bis\s+consistent\s+with\b|\bindicates?\b|\bimplies\b|
        \bmay\s+imply\b|\bcan\s+help\b|\bmay\s+help\b)
```

**Interpretation mapping.** HEDGE primarily reflects epistemic uncertainty markers (linked to $z_{\text{epistemic}}$). COND and ALT_CONTR are primary proxies for conditionality and contrastive alternative exploration (linked to $z_{\text{scope}}$). ADDITIVE is retained only as an exploratory discourse-marker count. DEFINITE (formerly ASSERT) is a coarse exploratory proxy for premature closure / definitiveness (linked to closure-proneness; often co-varying with $z_{\text{epistemic}}$).

### G.3 Optional structural indicators (language-agnostic)

The following patterns can be used as low-cost proxies for formatting and breadth (useful for spot-checking or robustness):

```
LIST_ITEM = (?m)^\s*(?:[-*• ]|[0-9]+[.)]|[①-⑳])+
JP_SENT_DELIM = [。！？!?]+
EN_SENT_DELIM = [.!?]+
EN_WORD = (?i)\b[0-9A-Z]+\b
```

**Recommended minimal reporting.** For each topic, report A/B deltas of the primary lexical endpoints (HEDGE_EPI, HEDGE_GEN, COND, ALT_CONTR); report ADDITIVE, DEFINITE, PROPENSITY, and LIST_ITEM count as exploratory. Because we use paired A/B comparisons with all non-linguistic variables held constant, these features are used as within-language deltas rather than absolute cross-language magnitudes.

## H   Worked Example of Forward and Inverse Use

This appendix provides a minimal worked example illustrating how the Z-model can be used in both forward (reporting) and inverse (diagnostic) modes. The purpose of this example is purely illustrative: it demonstrates how prompt-induced behaviors can be interpreted and adjusted using the Z-model, without claiming generality, optimization, or performance improvement.

### H.1   Scenario

We consider a short Japanese prompt fragment that is commonly used in practice and is known to elicit polite but relatively definitive responses. This example is intentionally minimal, focusing on a single fragment-level perturbation rather than a full prompt design.

### H.2   Forward observation

**Prompt (JP).**

   慎重に検討してください。

**Observed output (excerpt).**

> この方法は有効です。一般的に大きな問題はありません。

Although the prompt explicitly requests caution, the output exhibits categorical assertions with limited qualification. In particular, conditions, exceptions, or alternative cases are not made explicit.

### H.3 Inverse diagnosis via the Z-model

Based on the observable output characteristics, we report the following observer-side configuration using the Z-model. The polite phrasing and confident tone suggest elevated stylistic politeness ($z_{\text{style}} \uparrow$) together with increased epistemic commitment ($z_{\text{epistemic}} \uparrow$), while the absence of conditions or alternatives indicates suppressed conceptual breadth ($z_{\text{scope}} \downarrow$).

Importantly, these shifts are interpreted as coupled effects: the fragment primarily encodes an epistemic stance, but induces secondary effects on scope and closure through pragmatic coupling.

### H.4 Prompt adjustment (inverse to forward)

Suppose the user's intent is to retain a cautious tone while reducing premature definitiveness and encouraging explicit conditionalization. Based on the inverse diagnosis above, the prompt can be minimally adjusted to target lower epistemic commitment and broader scope.

**Adjusted prompt (JP).**

> 慎重に検討してください。断定せず、条件や例外があれば明示してください。

### H.5 Forward effect after adjustment

**Observed output (excerpt).**

> この方法は多くの場合に有効ですが、条件によっては例外も考えられます。

Relative to the original output, the adjusted response explicitly marks uncertainty, introduces conditional structure, and broadens the scope of consideration, while maintaining an overall cautious tone.

### H.6 Discussion

This worked example illustrates how the Z-model supports both forward reporting (prompt fragment $\rightarrow$ observable tendencies) and inverse diagnosis (observed behavior $\rightarrow$ prompt adjustment). Crucially, this interaction cannot be captured by a static taxonomy of prompt attributes: the same fragment simultaneously affects multiple observer-side axes, and effective adjustment requires reasoning about their coupled shifts. The example is not intended as an evaluation or optimization procedure, but as a concrete demonstration of the operational use of the Z-model.

To further mitigate model/snapshot dependence, we also report an open-weight model-sensitivity comparison on a pinned checkpoint with matched prompts and proxy definitions (Appendix J.2).

# I   Reproducibility Checklist and API Settings

**What must be recoverable under protocol re-execution.**   Because the paper evaluates a stochastic, protocol-level phenomenon, the main protocol re-execution target is recovery of the reported *directional and distribution-level* effects under matched prompts and counting rules. Exact string equality is neither expected nor required. A faithful protocol re-execution should recover the sign and qualitative ordering of the main API-window proxy deltas under the same model setting, while tolerating moderate changes in magnitude across model snapshots, hardware stacks, or open-weight substitutes. Cross-model differences, such as the open-weight absence of the Japanese COND increase, are treated as model-dependence diagnostics rather than failures of the API-window result.

**Embedded artifact set.**   Protocol re-execution requires only a compact artifact set: (i) the exact prompt templates and probe fragments, (ii) the model identifier and, for API-served runs, the pinned snapshot string and execution window, (iii) decoding parameters and sample counts, (iv) Unicode normalization and regex-based proxy definitions, and (v) the aggregation procedure used to compute deltas and bootstrap intervals. These elements are embedded in the paper and appendices rather than delegated to a large external software package: the protocol is specified in Sections C and 3.4, the indicator dictionary in Appendix G, and the model/API checklist in this section. No fine-tuning, learned classifier, or hidden annotation layer is required to recompute the reported measurements.

**Reader annotation and submission metadata.**   The artifact bundle includes anonymized reader-study materials. For the additional R5 form whose direct-reading eligibility was not verified, the bundle retains only an exclusion note; the raw unverified form is not included and no analysis depends on it. Because the manuscript reports human reader judgments, the OpenReview/TMLR human-subjects metadata should be updated consistently with the applicable institutional or company policy; the manuscript itself does not make an IRB/legal determination.

**API comparison path.**   The following snippet can be used to report the API-based setup concisely (values as in Section 3.4):

> *Model:* OpenAI `gpt-5.2-2025-12-11` (GPT-5.2 API snapshot; reported alias GPT-5.2).
> *Execution window:* 2026-02-27 UTC.
> *Sampling:* temperature $= 0.7$, top\_p $= 1.0$, max\_tokens $= 512$, $n = 50$ generations/(topic, condition) (five topics $\Rightarrow N = 250$ per condition).
> *Seed:* not set / not supported (stochastic sampling; evaluate via directional / distribution-level deltas).
> *Prompts:* system prompt fixed; user prompt fixed except for the probe fragment.
> *Post-processing & metrics:* Unicode normalization (NFKC), regex dictionary in Appendix G, bootstrap 95% CI.

**Supplementary audience-framing comparison path.**   For Exp. C (Section 3.5; Appendix F), use the same pinned OpenAI API snapshot `gpt-5.2-2025-12-11` and decoding parameters, with execution window 2026-05-23–24 UTC. The design uses five topics, two languages, two audience conditions (neutral vs. novice/non-specialist), and $n = 50$ generations per topic-condition, for 1000 generations total. The prompt is fixed except for the audience-framing fragment, and outputs are post-processed with NFKC

normalization and the reasoning-presentation proxies in Appendix F. The artifact path is `data/exp_c_framing_reasoning/raw/runs.jsonl`; run-level API provenance is recorded in `data/exp_c_framing_reasoning/raw/run_metadata.json` and summarized in `metadata/api_run_metadata.json`. Running `scripts/make_exp_c_from_raw_jsonl.py` regenerates the response-level proxy counts, the topic-aware bootstrap summaries, the data-quality report, and the Exp. C dot plot.

**Snapshot-stable comparison path.** Readers who prefer a non-API comparison route can use the open-weight setup in Appendix J, which pins `Qwen/Qwen2.5-7B-Instruct` to revision `a09a35458c702b33eeacc393d103063234e8bc28` while keeping the prompts, decoding settings, and observable proxy definitions matched to the API-based protocol.

The protocol in Section 3.4 and the dictionaries in Appendix G are intended to make protocol re-execution possible across contemporary LLMs, even if the precise API model snapshot is no longer available. For the main politeness probe, the corresponding run-level provenance file is `data/main_politeness/raw/run_metadata.json`; the bundle-level API metadata index is `metadata/api_run_metadata.json`. The raw JSONL keeps legacy combined ALT fields, while the paper-level ALT_CONTR/ADDITIVE split is canonicalized in the derived response-level feature table.

## J  Open-Weight Model-Sensitivity Comparison

To assess model and snapshot sensitivity of the reporting protocol, we apply (i) the cross-lingual politeness probing experiment (Exp. A) and (ii) the minimal directional sanity check (Exp. B) on a pinned open-weight, instruction-tuned checkpoint. Specifically, we use the model `Qwen/Qwen2.5-7B-Instruct`, revision `a09a35458c702b33eeacc393d103063234e8bc28`, obtained from the Hugging Face Hub. All prompts, decoding parameters, and proxy definitions are kept identical to the API-based setup; only the underlying model is changed.

### J.1  Cross-lingual Politeness Deltas (Exp. A)

Positive values indicate higher proxy rates under the polite condition than under the plain condition. The open-weight comparison is kept as a snapshot-stable model-sensitivity direction check; it is not used to expand the primary claims beyond the API-served pilot protocol. The result is intentionally mixed: the English HEDGE_EPI direction is positive on the pinned checkpoint ($\Delta = +0.273$, 95% CI [$+0.139, +0.402$]), whereas the Japanese COND direction is essentially absent ($\Delta = +0.003$, 95% CI [$-0.485, +0.470$]). This asymmetry is treated as an informative model-dependence result rather than as a confirmatory cross-model effect: JP COND is a robust API-window surface-marker shift, but not a model-family-invariant semantic effect.

Table J.1: Polite–plain deltas for the revised primary endpoint family on a pinned open-weight, instruction-tuned checkpoint. Bootstrap 95% confidence intervals are shown in brackets. This table is a snapshot-stable model-sensitivity check for the revised proxy definitions (HEDGE_EPI, HEDGE_GEN, COND, ALT_CONTR); DEFINITE and ADDITIVE remain exploratory and are not used as primary open-weight claims.

| Metric | $\Delta_{\mathrm{JP}}$ | $\Delta_{\mathrm{EN}}$ |
|---|---|---|
| HEDGE_EPI | +0.185 [-0.104, +0.529] | +0.273 [+0.139, +0.402] |
| HEDGE_GEN | -0.275 [-0.696, +0.063] | -0.094 [-0.214, +0.020] |
| COND | +0.003 [-0.485, +0.470] | +0.517 [+0.363, +0.664] |
| ALT_CONTR | -0.006 [-0.091, +0.088] | +0.049 [+0.002, +0.111] |
| DEFINITE (expl.) | +0.452 [+0.115, +0.776] | +0.024 [-0.115, +0.222] |
| ADDITIVE (expl.) | -0.067 [-0.253, +0.118] | +0.014 [-0.025, +0.059] |
| LIST_ITEMS | -1.368 [-3.360, +0.576] | -5.860 [-8.549, -2.852] |

## J.2 Directional Sanity Check (Exp. B)

Table J.2: Open-weight model-sensitivity comparison for the minimal sanity check (Exp. B; cf. Table C.1). We use the same Japanese base prompt and probe fragments as in Appendix C; English glosses are provided in parentheses for readability. Symbols indicate the direction of change in the chosen observable proxy relative to the base prompt on the pinned open-weight model (as in Table C.1). **Importantly, "+" and "–" denote changes in the observable proxy, not in the observer-side $z$-axis itself.**

| Prompt fragment | Target $z$-dimension | $\Delta$ ($T = 0.2$) | $\Delta$ ($T = 0.8$) |
|---|---|---|---|
| 箇条書きで (Bullets) | $z_{\mathrm{format}}$ | + | + |
| ステップごとに (Stepwise) | $z_{\mathrm{process}}$ | + | − |
| 断定せずに (Hedging) | $z_{\mathrm{epistemic}}$ | + | + |
| 慎重に (Cautious tone) | $z_{\mathrm{epistemic}}$ | + | + |
| A と B を比較して (Compare A/B) | $z_{\mathrm{process}}$ | + | + |
| 代表例を挙げて (Examples) | $z_{\mathrm{scope}}$ | + | + |

Table J.2 applies the Table C.1 protocol to the pinned open-weight model (same Japanese base prompt and probe fragments; English glosses shown for readability). We observe agreement in direction for 11/12 (6 fragments × 2 temperatures) entries; the only mismatch occurs for the stepwise fragment at $T = 0.8$, where the chosen minimal process proxy (e.g., explicit list-item markers) decreases relative to the base prompt. This likely reflects higher-variance formatting under high-temperature sampling (e.g., steps expressed without explicit bullet/number markers) rather than a qualitative reversal of procedural structuring.

## K Indicator-Set Ablation for Minimal Probing

A frequent reviewer concern for proxy-based probing is brittleness: do conclusions depend on a particular indicator choice? To address this without additional model calls, we perform a simple *indicator-set ablation* on the same generations used for Tables D.1 and D.4. We evaluate three nested indicator sets: (i) **Full** (10 indicators; all metrics reported in Table D.1), (ii) **Reduced** (7 indicators; core lexical indicators plus length/structure), and (iii) **Minimal** (5 indicators; core lexical indicators plus length).

Table K.1: Indicator-set ablation for minimal probing. $k$ is the number of indicators in the set. $\overline{|d|}$ is the mean absolute Cohen's $d$ over included indicators. #CI$\neq 0$ counts indicators whose bootstrap 95% CI for $\Delta$ excludes zero. #robust further requires a nontriviality filter, i.e., $|d| \geq \tau_d$ or JSD $\geq \tau_{\mathrm{JSD}}$ with $(\tau_d, \tau_{\mathrm{JSD}}) = (0.2, 0.05)$ (the revision-specified reporting rule in Section 4.5). Reduced and minimal sets retain the core lexical indicators (HEDGE, DEFINITE, COND, ALT_CONTR; ADDITIVE is exploratory) and show that the Japanese redistribution signature remains detectable under substantially fewer observables.

| Indicator set | $k$ | $\overline{|d|}_{\mathrm{JP}}$ | #CI$\neq 0$/#robust (JP) | $\overline{|d|}_{\mathrm{EN}}$ | #CI$\neq 0$/#robust (EN) |
|---|---|---|---|---|---|
| Full (10 indicators) | 10 | 0.363 | 8/7 | 0.259 | 6/6 |
| Reduced (7 indicators) | 7 | 0.478 | 7/6 | 0.263 | 4/4 |
| Minimal (5 indicators) | 5 | 0.355 | 5/4 | 0.126 | 2/2 |

Table K.2: Macro-group coarsening robustness check. Indicators from Tables D.1–D.4 are grouped by the four macro-groups of Figure 1. We report mean absolute Cohen's $d$ within each macro-group and the number of indicators whose bootstrap 95% CI for $\Delta$ excludes zero (#CI$\neq 0$), along with the number that additionally pass the nontriviality filter (#robust; this appendix-level legacy summary is not used for the primary claims).

| Macro-group | $k$ | $\overline{|d|}_{\mathrm{JP}}$ | #CI$\neq 0$/#robust (JP) | $\overline{|d|}_{\mathrm{EN}}$ | #CI$\neq 0$/#robust (EN) |
|---|---|---|---|---|---|
| Expression (format/length) | 3 | 0.830 | 3/3 | 0.514 | 3/3 |
| Epistemic | 5 | 0.137 | 3/2 | 0.201 | 3/3 |
| Reasoning (scope) | 2 | 0.231 | 2/2 | 0.018 | 0/0 |
| Framing (role/audience/objective) | 0 | | – | | – |

For each set and language, we summarize (a) the mean absolute standardized effect size $\overline{|d|}$ across included indicators, and (b) the number of included indicators whose bootstrap 95% CI for $\Delta$ excludes zero. Under the matched probing protocol, Japanese often exhibits a more multi-faceted proxy signature than English; we treat this as a protocol-level diagnostic observation and note possible confounds from cross-lingual training/alignment differences.

**Macro-group coarsening (robustness under 4-group coarsening).** Beyond ablations over observable indicators, we also test robustness under coarsening of the $Z$-space itself by collapsing observables into the four macro-groups of Figure 1. We assign indicators from Tables D.1–D.4 to macro-groups as follows: **Expression** (structural/format proxies: `chars`, `list_items`, `sentences`), **Reasoning** (scope proxies: COND, ALT_CONTR, with ADDITIVE exploratory), and **Epistemic** (uncertainty/closure proxies: HEDGE_*, DEFINITE, PROPENSITY). The **Framing** macro-group (role/audience/objective) is not operationalized in Exp. A and is therefore not evaluated here. We include the intended **Expression** shift for completeness; secondary redistribution concerns the off-target macro-groups.

## L Additional Figures: P1 Orthogonal Design, 11D Coarsening, Basis Rotation, and 2×2 Factorial Probe

### L.1 P1 orthogonal 4D design: stance targets (target-wise view)

Figure 2 reports mean $R^2$ aggregated over (i) all proxy targets and (ii) stance-only targets. Figure L.1 breaks the stance-only aggregate down by target to show which stance proxies benefit from additional dimensions.

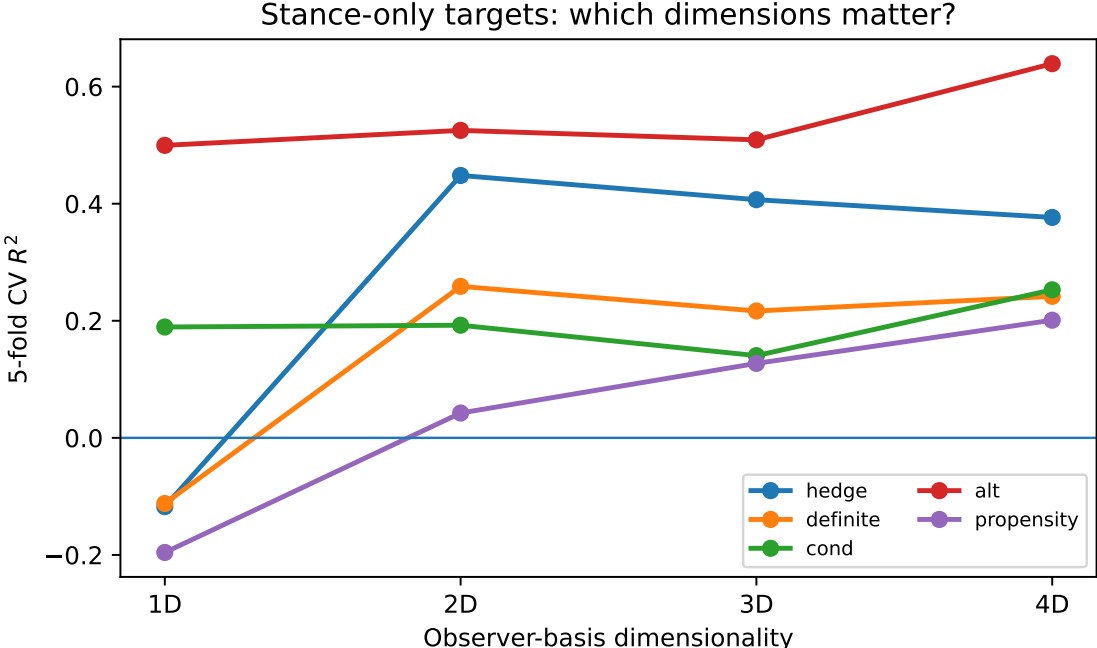

Figure L.1: **P1 orthogonal 4D design: target-wise stance descriptive fit under basis reduction.**
5-fold cross-validated $R^2$ for each stance proxy target under reduced observer bases (1D–4D). This target-wise
view complements Figure 2 and shows that the $z_{\text{epistemic}}-z_{\text{scope}}$ subspace explains a substantial portion of
stance variation, while additional dimensions provide more modest gains that differ by target.

## L.2  11D coarsening check on the cross-lingual politeness probe (task-sliced results)

Beyond the controlled 4D orthogonal design above, we also test basis coarsening for the full
11D reference basis on the cross-lingual politeness probe (Exp. A). Figure L.2 provides a
task-sliced view (task0/task1) under the same proxy-delta descriptive-fit protocol.

## L.3  Basis rotation robustness

To probe robustness to alternative parameterizations beyond coarse merging/splitting, we
apply random orthogonal rotations within each macro-group subspace of the observer basis and
re-evaluate the same proxy-delta descriptive-fit protocol. Figure L.3 shows that performance
is stable under such rotations, consistent with redistribution detection being insensitive to
axis labeling within the same representational subspace.

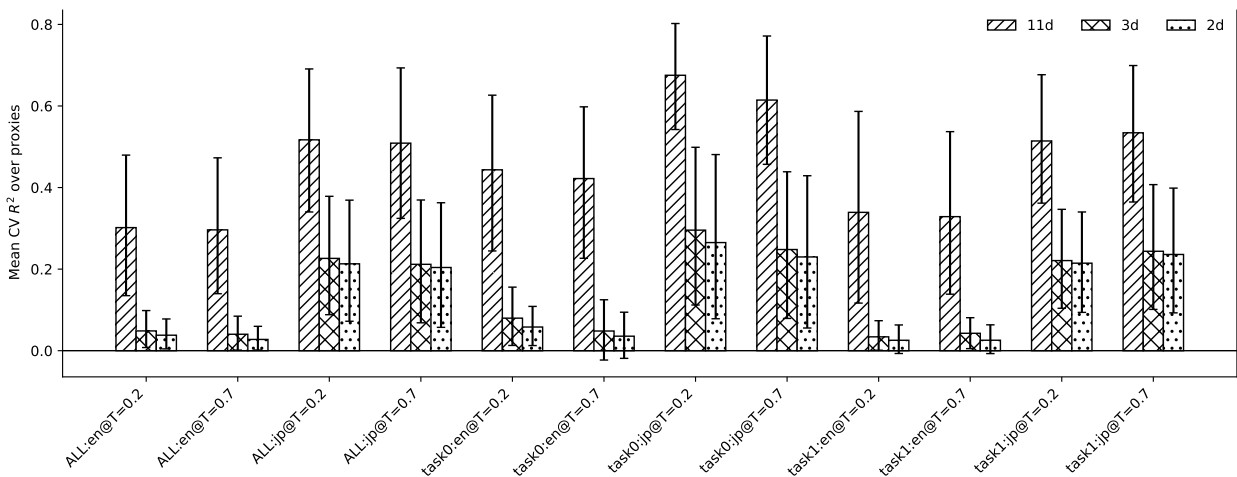

Figure L.2: **11D coarsening check by task slice (monochrome-friendly).** Mean cross-validated $R^2$ over proxy metrics when fitting proxy-delta signatures from the observer basis coordinates, comparing the full 11D reference basis to reduced 3D/2D coarsenings, shown for the task-averaged slice and for each task separately. This plot illustrates protocol- and proxy-dependent reporting-resolution differences rather than validating an intrinsic 11D dimensionality.

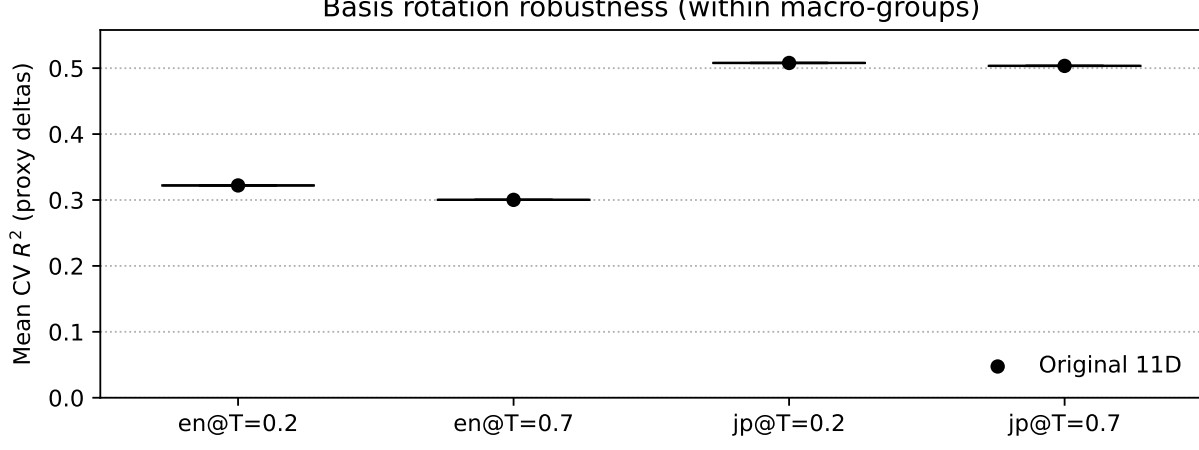

Figure L.3: **Basis rotation robustness (within macro-groups; monochrome-friendly).** Mean cross-validated $R^2$ for fitting proxy-delta signatures from the observer basis coordinates under random orthogonal rotations within each macro-group subspace (200 rotations; bands show the 2.5/97.5 percentiles across rotations). The stability indicates that redistribution detection is not an artifact of a specific axis labeling.

### L.4   Factorial 2×2 probe: cell means for representative proxies

This appendix reports the ALL-slice factorial summaries used in the figures below. The raw data comprise one representative topic, two task templates (task0/task1), four factorial conditions, and 30 stochastic generations per task–condition cell. The plotted cell means and interaction summaries pool the two task templates, so each displayed condition contains 60 generations. Task-specific slices are retained only as audit material and are not the basis of the figures reported here.

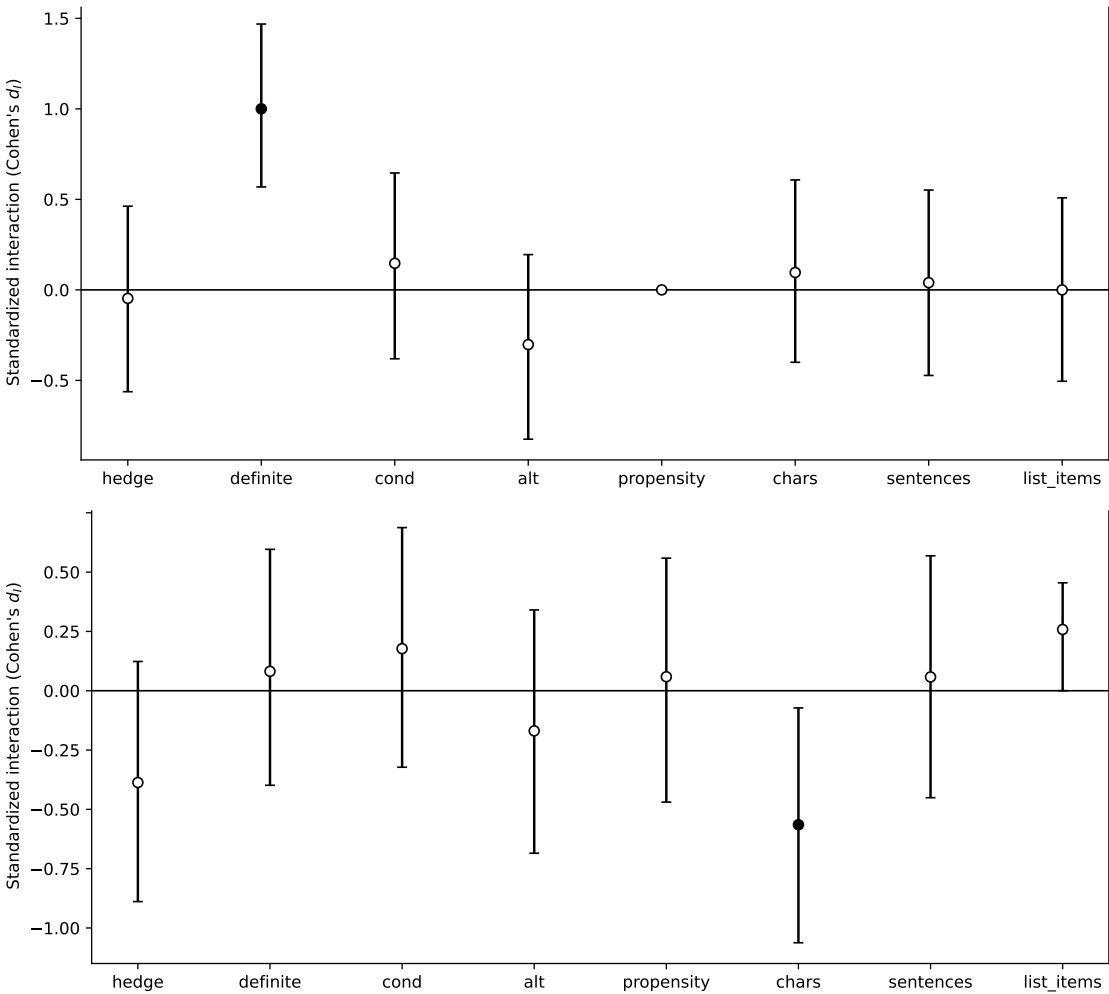

Figure L.4: **Exploratory factorial** $2 \times 2$ **probe: standardized interaction effects (monochrome-friendly).** Standardized interaction effect size $d_I$ with 95% bootstrap confidence intervals for each proxy metric, shown separately for English (top) and Japanese (bottom). The plotted ALL slice pools two task templates: each task–condition cell has 30 generations, yielding $n = 60$ outputs per condition after pooling. The probe is included as an exploratory diagnostic on one topic, not as primary evidence.

Figure L.4 summarizes interactions as standardized effect sizes. Figure L.5 visualizes raw cell means for representative proxies in each language, together with the additive expectation for $C11$ (dashed line), making non-additivity directly visible.

**Appendix L.4 bundle note.** In the reviewer bundle, the canonical factorial artifacts for Appendix L.4 are shipped as precomputed ALL-slice outputs under the following two directories:

```
data/factorial/derived/figY_all/en_T0.2_ALL/
data/factorial/derived/figY_all/jp_T0.2_ALL/
```

These directories contain the underlying summary CSV and rendered cell-means figures for the ALL slice. They are derived from the canonical `T=0.2` raw JSONL files by pooling task templates with `-task_id -1` (30 generations per task–condition cell, 60 outputs per condition after pooling) and running:

```
scripts/factorial/run_figY_all.sh
# equivalently: scripts/factorial/make_figY_factorial_plots.py --task_id -1
```

Task-specific slices are retained only as audit/sensitivity material; the paper figures use the ALL slice. Regeneration is therefore optional for reviewer inspection. The bundle also retains an additional Japanese `T=0.7` diagnostic raw run from an earlier exploratory sweep. That extra Japanese-only run is not used in Figures L.4 and L.5 or in any reported factorial summary, and no English `T=0.7` factorial run was collected for this exploratory check.

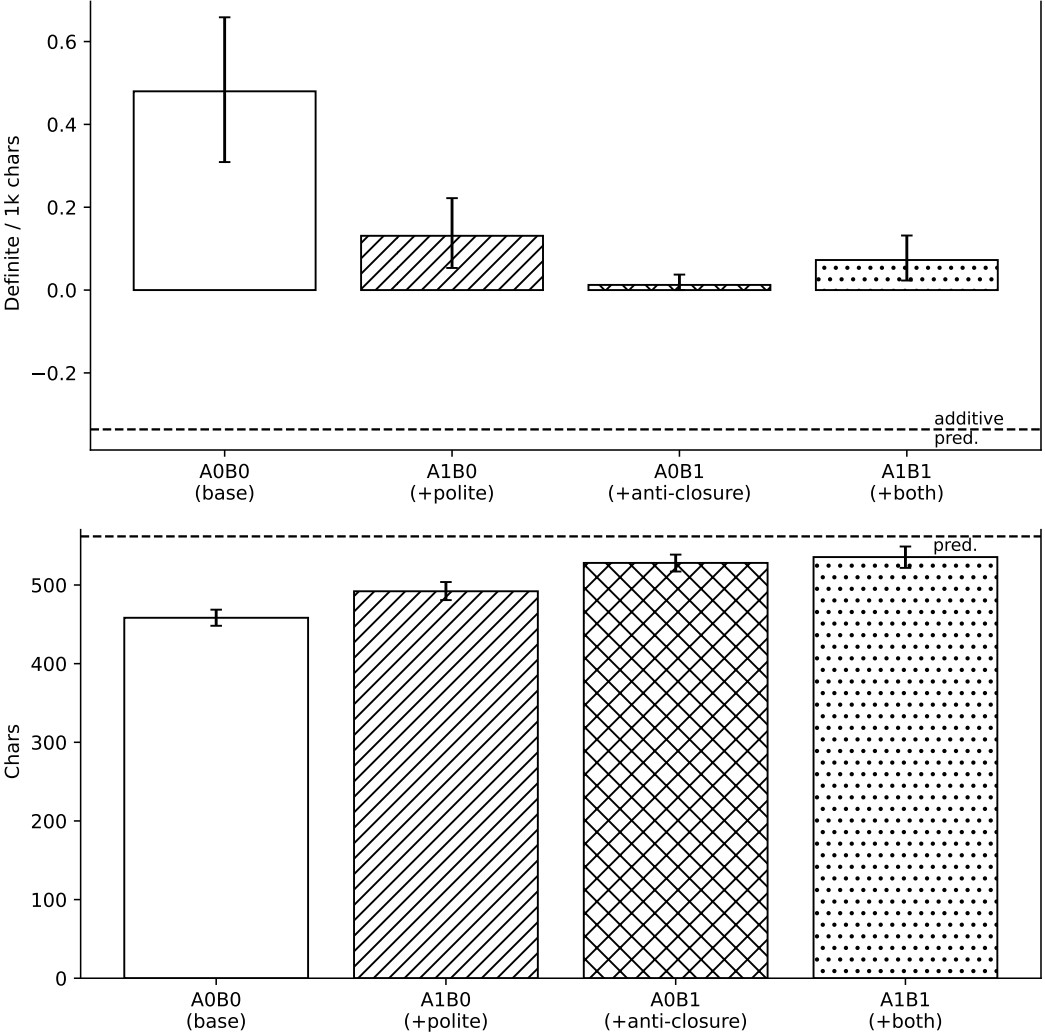

Figure L.5: **Factorial 2×2 probe: raw cell means with additive expectation (monochrome-friendly).** Means (with 95% bootstrap confidence intervals) for representative proxies under the four factorial conditions $C00 = p_0$, $C10 = p_0 \oplus A$, $C01 = p_0 \oplus B$, $C11 = p_0 \oplus A \oplus B$, using the ALL slice that pools two task templates (30 generations per task–condition cell; $n = 60$ per condition after pooling). The dashed line denotes the additive expectation for $C11$ under no interaction ($\mu_{10} + \mu_{01} - \mu_{00}$). Deviations from the dashed line correspond to the interaction term summarized in Figure L.4.

## M  Optional probabilistic view: mixture over observer-side configurations

Let $x$ denote a prompt and $y$ a generated output. One can view the observer-side configuration as a reporting variable $z$ with a prompt-dependent distribution $p(z \mid x)$, and write the output distribution as a mixture:

$$p(y \mid x) \;=\; \int p(y \mid z)\, p(z \mid x)\, dz. \tag{M.1}$$

This interpretation is intended purely as an alternative *reading* of the observer-side abstraction. It does not introduce an estimator for $z$ and is not used in the empirical procedures, which remain grounded in observable proxy measurements $r(y)$ under controlled A/B perturbations.

