# OpenReview forum: "Observer-Side Diagnosis of Prompt-Induced Behavioral Shifts in Large Language Models: A Diagnostic Vocabulary and a Pilot Japanese/English Politeness Probe"
_TMLR — Under review for TMLR_

### Review · Reviewer_KpqH · 2026-05-23

**Summary Of Contributions:**

This paper investigates "latent interference" in Large Language Models, the phenomenon where a prompt fragment with changes to a single surface property (e.g., style or politeness) unintentionally triggers shifts in secondary behaviors of a LLM model, such as epistemic stance (confidence/hedging), reasoning presentation, and conceptual scope.

To systematically describe and diagnose these shifts, the authors propose the Z-model, an 11-D observer-side diagnostic vocabulary $z(p)$ categorized into four macro-groups (Framing, Reasoning, Expression, and Epistemic control). To track behavioral effects, the paper uses lightweight, surface-level lexical proxies (regex counting) $r(y)$ for diagnosis. The authors employ a "minimal cross-lingual probe" using Japanese as a typological stress test. Because Japanese grammatically bundles politeness with social and epistemic stance, the authors use it to demonstrate how a single stylistic cue causes complex, multi-dimensional redistribution of uncertainty markers, contrasting this with more straightforward additive effects seen in English.

**Audience:**

Yes

**Audience Explanation:**

Yes. Framing prompt effects as observer-side coupled shifts across multiple behavioral dimensions is a useful angle to evaluate LLMs. ML engineers and researchers, particularly those working on multilingual aspects of LLM, will be interested in the framework.

**Broader Impact Concerns:**

No concern on the ethical implications

**Claims And Evidence:**

Yes

**Claims Explanation:**

The authors have been very disciplined about scoping their claims. Two gaps worth to mention: 1) only one cross-group pathway is empirically tested (Framing dimensions are entirely unprobed), and 2) the empirical conclusions are built on top of regex-based lexical proxies whose construct validity is not established against human judgment. The conceptual scaffold is therefore better-evidenced as a diagnostic framework than as a comprehensive theory, which is how the authors position it.

**Requested Changes:**

Major Changes:

1. Demonstrate at least one additional cross-group pathway. The empirical validation is currently limited to a single pathway (politeness to epistemic/scope shifts). A single additional probe - for example, a Framing-side cue (persona or audience) and its secondary effects on Reasoning proxies - would substantially strengthen the claim that the macro-group structure is a useful diagnostic vocabulary rather than a scaffold built around one observation.

2. Validate the lexical proxies against semantic judgment. The empirical foundation is based on regex-based counts as proxies for epistemic stance and scope. The authors should include a small-scale human evaluation on a subset of the data (e.g., 100 paired responses) to verify that the lightweight regex counts actually correlate with human perceptions of confidence, hedging, and scope.  If full human evaluation is infeasible, extending the LLM-judge design from Appendix A.5 to the main probe - with correlations between judge ratings and lexical proxies reported per condition - would be a weaker but useful alternative.

Minor Changes:

3. Tables 3–6 are dense and hard to read. the point estimates and confidence intervals make it impossible to properly characterize the statistical significance of the results. These tables need to be reformatted for clarity. Consider to convert the tables into plots (as Figure 4 already does for the factorial probe).

4.  Table 1, the numbered list in Section 3.2, and Table A.1 are three near-redundant versions of the same content. Consolidate them into a single reference table.

5.  Japanese is central to the argument but many phrases appear without English translations. The authors should add literal English translations after every Japanese fragment in the main text and tables.

6. Restructure Section 4 for clarity. The core methodological content - forward and inverse procedures - is fragmented across four subsections (4.3–4.6) and scattered by linguistic background (4.1), phenomenon reinterpretation (4.7), Bayesian reading (4.8) and others. Consider to restructure the section for better readability.

---

> ### Author Response · Authors · 2026-05-23
> **Response to Reviewer KpqH (Revision in Progress)**
>
> Dear Reviewer KpqH,
>
> We sincerely thank you for your thoughtful review and highly
> constructive feedback. We are grateful that you recognized the
> intended scope of the Z-model as an observer-side diagnostic
> framework.
>
> We agree with both major points raised in your review: the need
> to demonstrate an additional cross-group pathway, and the need to
> validate the lexical proxies against semantic judgments. We also
> appreciate your helpful suggestions regarding table readability,
> consolidation of redundant material, translations of Japanese
> examples, and the restructuring of Section 4.
>
> We are currently conducting additional API-based probing
> experiments and a small-scale human evaluation to address these
> points more directly. In the revised manuscript, we will
> incorporate the new empirical results and provide a detailed
> point-by-point response to each of your comments.
>
> Thank you again for your constructive guidance, which will help
> us substantially strengthen the paper.
>
> Sincerely,
> The Authors

---

> > ### Author Response · Authors · 2026-06-08
> > **Response to Reviewer KpqH: added cross-group pathway (Exp. C) and proxy–semantic checks**
> >
> > We thank the reviewer for the supportive and precise review, and for identifying two gaps: the earlier version tested only one cross-group pathway, and the construct validity of the automatic proxies needed clearer evidence and framing. The revision addresses both points while retaining the pilot scope endorsed by the review.
> >
> > **Major 1 — Additional cross-group pathway.** We added Exp. C in Section 3.5, Figure 7, and Appendix F. The manipulation targets z_audience: a neutral explanation prompt is contrasted with a matched novice/non-specialist audience cue. The English cue is "for a non-specialist reader," and the Japanese cue is *shogakusha ni mo wakaru you ni*, "so that even a beginner can understand." The experiment is recomputed from the raw 5-topic × 2-language × 2-audience-condition × 50-generation dataset. Novice/non-specialist framing increases analogy-oriented and explanatory-scaffolding markers in both languages, while conditional markers do not uniformly increase. We therefore present Exp. C as a second audited path and support layer, not as a second primary endpoint family or as broad validation of the Framing or Reasoning macro-groups. Appendix F reports the full prompts, dictionary, raw-count summaries, stricter analogy-dictionary robustness check, and data-quality report.
> >
> > **Major 2 — Validating proxies against semantic judgment.** We implemented both the human check and the LLM-judge alternative suggested in the review. The direct-reader check is condition-blind and uses a balanced subset of 200 outputs, analyzed by direct-reading eligibility for each language (Appendix E, Figure 6). We also added a rubric-based LLM-judge diagnostic over the full main politeness probe: Table E.7 reports condition-stratified judge–proxy Spearman correlations over all 1000 outputs. Because inter-reader agreement is low and the proxy–judgment correlations are mixed, we report these layers as exploratory transparency rather than validation. Section 4.2 explicitly frames the divergence as a construct-validity issue: automatic proxies are transparent indicators of surface-marker redistribution, not calibrated semantic measurements. This keeps the empirical claim at the level of observable redistribution, consistent with the reviewer's reading of the framework as a diagnostic vocabulary rather than a comprehensive theory.
> >
> > **Minor.** Dense tables are replaced in the main text by forest-style CI plots and concise takeaways, with numerical detail moved to Appendix D. The Section 3.2 prose enumeration was removed so that Table 1 is the single canonical inventory, with Table B.1 retained as an appendix reference sheet. Japanese fragments now carry romanization and literal glosses where they appear in the main text, and Section 4 is reorganized into the five focused subsections described above.
> >
> > We are grateful for the guidance, which materially strengthened both the scoping and the evidential framing of the paper.

---

> > > ### Comment · Reviewer_KpqH · 2026-06-13
> > >
> > > Thanks for the thorough and responsive revision. Both major requests have been addressed. The scope of the updated manuscript is now well aligned with the supporting evidence as well.
> > >
> > > For major 1: The new Exp. C (Section 3.5, Figure 7, Appendix F) implements the requested probe with matched size and protocol as the main probe. The finding is appropriately scoped as well - analogy/scaffolding markers increase in both languages while conditional markers do not uniformly increase. Note that Exp. C ran in a later API window than the main probe, so it does not show that a single model state supports both pathways simultaneously; this is a fair limitation rather than an outstanding issue.
> > >
> > > For major 2: The authors implemented both suggested paths: a human study (Appendix E, Figure 6) and a full-sample LLM check (Table E.7). Note that the validation itself is inconclusive (e.g. zero inter-reader agreement), and the authors reframed the contribution to be consistent with the results.
> > >
> > > Minor requests are addressed properly as well.
> > >
> > > Overall, the new manuscript strengthened the scope of the paper and consolidated the framing. I appreciate the authors' careful revision.

---

> > > > ### Author Response · Authors · 2026-06-14
> > > > **Reply to Reviewer KpqH**
> > > >
> > > > We thank Reviewer KpqH for the careful re-reading and for confirming that the revised scope is aligned with the supporting evidence. Both points are stated as explicit limitations in the revised manuscript (Sections 3.4–3.5; Appendices E–F), and the contribution has been scoped accordingly. We appreciate the reviewer's constructive engagement throughout the discussion period.

---

### Review · Reviewer_4w4A · 2026-05-31

**Summary Of Contributions:**

This paper proposes an observer-side diagnostic vocabulary for describing prompt-induced behavioral shifts in LLMs. The framework groups prompt effects into four macro-groups, framing, reasoning, expression, and epistemic control, and instantiates them as an 11-axis "Z-model" intended for black-box analysis rather than model-internal interpretation. Empirically, the paper studies a narrow Japanese/English stress test: whether a politeness cue changes surface proxies for uncertainty, definitiveness, conditionality, and alternatives. The authors report bootstrap confidence intervals for a GPT-5.2 API run, a small 2x2 factorial probe, and a partial reproduction on Qwen2.5-7B-Instruct.

**Additional Comments:**

### Weaknesses

The paper is not ready for publication in its current form. The main problems are clarity, excessive length, weak empirical grounding for the broad framework, and poor organization of tables/figures.

1.  **The conceptual contribution is overextended relative to the evidence.** The paper introduces a large amount of new terminology, including macro-groups, the Z-model, latent interference, forward mapping, inverse diagnosis, observer-side latent blocks, basis reduction, coarsening, and rotation. However, the empirical study only tests one narrow pathway: the effect of a politeness cue ("politely"/"丁寧に") on a few lexical proxies. This does not convincingly validate an 11-axis diagnostic framework. At most, the current evidence shows that one style-oriented prompt fragment can change some surface markers under one specific protocol. The mathematical notation, such as latent vectors, encoder/decoder maps, and inverse mappings, gives the appearance of a formal model but does not correspond to an identifiable estimator, causal model, or predictive method. The paper should either provide substantially stronger evidence for the full framework or narrow the contribution to a focused pilot study of politeness-induced proxy shifts.

2.  **The empirical design and statistical analysis are too weak for the claims.** The main experiment uses only five benign technical topics, one primary cue, one API model, and one open-weight reproduction model. The factorial 2x2 probe is even narrower, using a single topic with n = 50 generations per cell. This is insufficient to support broad claims about structured latent interference, Japanese prompting, or language-dependent diagnostic behavior. The statistical treatment also needs improvement. The bootstrap appears to treat generations as independent samples even though the design is nested by topic and condition; a topic-aware or hierarchical bootstrap would be more appropriate. The paper reports many proxy comparisons without clear multiple-comparison handling or pre-specified primary metrics. Several results are marginal, mixed, or proxy-dependent, yet the narrative often interprets them as a coherent interference signature.

3.  **The proxy measurements are fragile and insufficiently validated.** The core measurements are regex counts of lexical markers per 1,000 characters. These are shallow proxies for complex constructs such as epistemic stance, scope, alternatives, and hallucination risk. For example, the ALT proxy includes additive discourse markers such as "also" and "furthermore", which do not necessarily indicate genuine alternative reasoning; HEDGE_GEN conflates genericity, typicality, and uncertainty; and ASSERT does not reliably measure categorical tone. The length normalization is also potentially problematic because prompt politeness itself changes output length, so per-character rates may introduce artifacts. Most importantly, there is no human annotation showing that the proxy shifts correspond to perceived uncertainty, definitiveness, politeness, scope, or reliability. Without such validation, the empirical results should be framed as surface-level discourse-marker shifts, not as evidence of safer or less reliable model behavior.

4.  **The writing, layout, and accessibility of the Japanese material are major problems.** The manuscript is 49 pages total, with roughly 30 pages before references. TMLR does not impose a strict page limit, but unusually long papers should justify their length; this paper is repetitive and does not make good use of its length. Many ideas are restated across the introduction, Section 3, Section 4, limitations, reproducibility statement, and appendices. The tables and figures are also poorly organized. Tables 2, 7, and A.2 mix Japanese text, English text, mathematical notation, arrows, and long explanations in cramped layouts, making them difficult to parse. Table 6 and Figure 4 consume substantial space while conveying limited information. Because Japanese is central to the paper, Japanese examples are necessary, but many prompt fragments and table entries lack consistent English translations, glosses, or romanization. This makes the work hard to evaluate for TMLR reviewers who are not Japanese speakers.

5.  **Novelty and reproducibility remain unclear.** The central observation that small prompt changes can affect style, reasoning, confidence, and scope is already well established in prompt sensitivity and behavioral testing literature. The paper needs to explain more sharply what the Z-model enables beyond a simpler prompt-feature taxonomy plus paired perturbation tests. Reproducibility is also incomplete: although the paper includes regexes and prompt descriptions, it does not appear to release raw generations, exact API request bodies, response metadata, analysis scripts, or the code used for counting and bootstrapping. The LLM-as-a-judge check uses another black-box model and is not a substitute for human validation or full artifact release.

**Audience:**

Yes

**Audience Explanation:**

N/A

**Claims And Evidence:**

No

**Claims Explanation:**

Please check the weakness section in the additional comment.

**Requested Changes:**

1.  **Narrow the claims** to match the evidence. Present the empirical result as a targeted politeness-probe study, not as broad validation of an 11-axis framework.

2.  **Shorten and reorganize the paper.** Move most examples, Japanese linguistic background, regex details, and extended tables to the appendix. Remove repeated explanation of the same limitations and scoping statements. Replace dense prose tables with compact summaries, provide translations/glosses for all Japanese text, reduce whitespace, and ensure each figure has a clear takeaway.

3.  **Strengthen empirical validation.** Add more prompt fragments, more topics, more model families, and more languages, or explicitly state that the contribution is a pilot diagnostic case study. Add human annotation or at least a carefully designed semantic evaluation showing that the regex proxies correspond to perceived uncertainty, definitiveness, scope, and alternative reasoning.

4.  **Improve statistical analysis.** Use topic-aware or hierarchical bootstrap, identify primary metrics in advance, report multiple-comparison handling, and avoid interpreting marginal proxy shifts as broad structured interference.

---

> ### Author Response · Authors · 2026-06-08
> **Response to Reviewer 4w4A: rescoped claims, topic-aware statistics, and added reader/LLM-judge checks**
>
> We thank the reviewer for the detailed and constructive review. We took the central concern to be that the original claims exceeded the evidence. The revision therefore treats claim narrowing, rather than claim preservation, as the appropriate remedy under TMLR's evidence-matching criterion. We also understand the review as already positive on audience interest, so the response below focuses on aligning the claims with the supported evidence.
>
> **R1 — Claim scope and overextension.** We agree and have rescoped the manuscript throughout. The title and abstract now foreground a diagnostic vocabulary plus a pilot Japanese/English politeness probe. The abstract, the opening of Section 3, the limitations, and the conclusion state explicitly that the work does not validate the full 11-axis framework or recover hidden model states. The Z-model is described as an observer-side reporting vocabulary and auditable reference basis, not as an estimator, causal model, or predictive control method. The compact mathematical block is now labeled as bookkeeping notation rather than as an identifiable model.
>
> **R2 — Length, organization, tables, glosses, and figure takeaways.** We reorganized the paper to reduce main-text density. Japanese linguistic background is moved to Appendix A, regex dictionaries to Appendix G, and numerical tables to Appendix D. The main text now uses forest-style CI plots and compact narrative summaries for the primary results, raw-count check, reader check, and supplementary audience-framing probe. Japanese examples retained in the main text and Table 2 now carry romanization and literal English glosses. Section 4 has been restructured into five focused subsections: what the pilot supports, construct validity, what the Z-model adds, the roles of Japanese and English, and the practical audit recipe. Repeated scoping statements were consolidated.
>
> **R3 — Empirical validation and proxy–perception correspondence.** We adopted both remedies suggested by the review, while keeping them within the pilot scope. First, we explicitly adopt a pilot-diagnostic framing, add a pinned open-weight comparison path on Qwen/Qwen2.5-7B-Instruct in Appendix J, and add a second audited cross-group pathway: Exp. C, Framing→Reasoning, in Section 3.5 and Appendix F. Second, we added two semantic-correspondence checks: a condition-blind direct-reader check on 200 balanced outputs (Appendix E, Figure 6) and a rubric-based LLM-judge diagnostic over all 1000 main-probe outputs (Table E.7; judge model `gpt-5-mini-2025-08-07`). We report transparently that inter-reader agreement is low, with QWK values near zero, and that proxy–perception correlations are mixed. We therefore do not claim that the proxies are calibrated semantic measures. Section 4.2 reframes this as a construct-validity issue: lexical indicators capture enumerative surface-marker redistribution, whereas reader judgments concern relevance-weighted perceived sufficiency. Consistent with the first acceptance criterion, the empirical claim is limited to within-protocol surface-marker redistribution; the reader and judge layers provide transparency rather than confirmatory validation.
>
> **R4 — Statistics.** We adopted the requested statistical changes. The revised primary analysis uses a topic-aware hierarchical bootstrap that resamples topics and then generations within each topic–condition cell. Before recomputing the revised tables and figures, we fixed a revision-specified four-endpoint primary family: HEDGE_EPI, HEDGE_GEN, COND, and ALT_CONTR. We describe this as revision-specified, explicitly not as preregistration. We report Benjamini–Hochberg FDR across the eight primary tests and add raw-count robustness checks to rule out length-normalization artifacts. We no longer interpret marginal shifts as a global interference signature. The robust effects are localized to English HEDGE_EPI and Japanese COND; Japanese COND is also flagged as not reproduced on the pinned open-weight checkpoint.
>
> **Minor.** The redundant prose enumeration in Section 3.2 was removed. Table 1 is now the single canonical main-text inventory, with a compact formal reference sheet retained only in the appendix. Japanese fragments are glossed, Section 4 is restructured as described above, and the appendix now contains the expanded materials needed for audit and reproduction.
>
> We hope these changes bring the claims into alignment with the evidence, and we would welcome any remaining scope statement that the reviewer still reads as overclaiming.

---

> ### Author Response · Authors · 2026-06-28
> **Follow-up to Reviewer 4w4A**
>
> **Follow-up to Reviewer 4w4A: status after the latest revision**
>
> We thank the reviewer again for the detailed initial review. Since our previous response, we have further revised the manuscript and supplementary bundle, while keeping the empirical claim deliberately narrow. This follow-up summarizes where the reviewer's main concerns are now addressed. Consistent with TMLR's first acceptance criterion, the remedy throughout is to match the claims to the evidence rather than to enlarge them.
>
> **1. Claim scope.** The manuscript is now explicitly framed as a diagnostic vocabulary plus a pilot Japanese/English politeness probe, not as validation of the full 11-axis framework. The abstract, Section 3, the limitations, and the conclusion state that the paper does not recover hidden model states or validate the full reference basis. We added a Claim-Evidence Boundary table (Table 3) clarifying what is supported, what is exploratory, and what is not claimed. The compact mathematical block (Eq. 1) is described as bookkeeping notation, not an identifiable estimator or causal model.
>
> **2. Z-model presentation.** The Z-model presentation has been softened and more tightly scoped. The main text now presents it as an observer-side reporting vocabulary rather than a latent-dimensional theory, and the stronger "latent interference" language has been replaced by "operational secondary redistribution" and "observer-side reporting axes." The 11-axis basis is retained only as a reporting resolution: Section 3.2, Figure 2, Appendix K, and Appendix L report basis-reduction, coarsening, rotation, and indicator-set checks, while explicitly disclaiming that 11 is minimal, unique, or geometrically meaningful.
>
> **3. Proxy definitions and fragility.** The proxy-based evidence is now scoped at the marker level. The earlier ALT proxy has been split into ALT_CONTR and ADDITIVE, with ADDITIVE treated as exploratory; ASSERT has been renamed DEFINITE and treated as exploratory; and HEDGE has been split into HEDGE_EPI and HEDGE_GEN. The primary endpoint family is revision-specified as HEDGE_EPI, HEDGE_GEN, COND, and ALT_CONTR. We also added raw-count checks for the primary findings to address the length-normalization concern.
>
> **4. Statistics.** The statistical analysis has been revised as requested. The primary analysis now uses a topic-aware hierarchical bootstrap, resampling topics and then generations within topic-condition cells. Benjamini-Hochberg FDR is reported across the eight primary tests. The robust effects are localized to English HEDGE_EPI and Japanese COND, and marginal or mixed shifts are no longer interpreted as a global structured-interference signature.
>
> **5. Proxy-perception correspondence.** This is now treated as a construct-validity boundary rather than as confirmed validation. We added a condition-blind direct-reader check on 200 balanced outputs and a rubric-based LLM-judge diagnostic over all 1000 main-probe outputs. Because inter-reader agreement is low and proxy-judgment correlations are mixed, the manuscript explicitly states that these layers provide exploratory transparency, not confirmatory semantic validation. Section 4.2 frames the automatic proxies as transparent indicators of surface-marker redistribution, not as estimators of reader-perceived semantics, factuality, safety, or calibrated uncertainty.
>
> **6. Reproducibility and artifact consistency.** The supplementary bundle now includes raw generations, response-level feature tables, counting/bootstrap scripts, the LLM-judge request/response logs and batch identifiers, run-level API provenance metadata, the pinned API snapshot identifier `gpt-5.2-2025-12-11`, anonymized reader-check materials, manifest/checksum files, and synchronized manuscript/bundle tables. The open-weight Qwen run is now described as a model/snapshot-sensitivity comparison rather than cross-model validation.
>
> In short, the revision follows the remedy requested in the review: it narrows the paper to a focused pilot study of surface-marker redistribution, improves the statistical treatment and artifact trail, and avoids using proxy shifts as evidence of broad semantic, safety, or 11-axis framework validation. If any specific scope statement still reads as overclaiming, we would be happy to address it.
>
> Sincerely,
> The Authors

---

### Review · Reviewer_4LQX · 2026-06-28

**Summary Of Contributions:**

Summary:

The paper introduces an observer-side diagnostic vocabulary to systematically describe how localized prompt fragments induce unintended, secondary behavioral shifts in LLM outputs. Rather than claiming to identify internal latent states, the authors operationalize this framework via a reproducible black-box A/B probing protocol that tracks lightweight surface markers. The primary empirical contribution is a cross-lingual pilot study demonstrating that a simple politeness cue systematically redistributes epistemic hedging in English and conditional framing in Japanese.

Strengths:

S1. The experimental design is highly controlled, utilizing topic-aware hierarchical bootstrapping and Benjamini-Hochberg FDR corrections to ensure robust, distribution-level statistical claims.

S2. The authors display commendable restraint by strictly differentiating mechanical enumerative breadth from human-perceived relevance-weighted sufficiency. They avoid the common pitfall of overclaiming latent interpretability.

S3. The diagnostic pipeline is transparent, providing exact regex dictionaries, clear prompt templates, and a supplementary open-weight replication check to verify API-based findings.

Weaknesses:

W1. The study heavily relies on shallow lexical and structural regex proxies. The validity of these proxies is undermined by the human reader check, which yielded near-zero inter-rater agreement, indicating that the mechanical increase of discourse markers does not reliably translate to human-perceived semantic shifts.

W2. The proposed 11-axis Z-model framework feels over-engineered considering the highly narrow empirical evaluation. The study only tests two isolated pathways on a limited set of 5 benign technical topics, making the broad 11-dimensional vocabulary seem largely speculative.

W3. The failure to replicate the robust Japanese conditional framing shift (found in the GPT-5.2 API) on the pinned open-weight Qwen model raises generalization concerns. This discrepancy suggests the observed latent interference might be a snapshot-specific alignment artifact rather than a general linguistic or pragmatic phenomenon.

**Audience:**

Yes

**Audience Explanation:**

The reviewer believes that researchers focused on LLM evaluation, alignment, and prompt engineering will find the paper's systematic methodology for auditing unintended, prompt-induced behavioral shifts relevant.

**Broader Impact Concerns:**

The authors have already provided a sufficient Broader Impact Statement that adequately addresses potential risks.

**Claims And Evidence:**

Yes

**Claims Explanation:**

While the statistical methodology for tracking surface-marker redistribution is rigorous, the evidence for actual semantic shifts is less convincing due to the reliance on shallow regex proxies and near-zero human inter-rater agreement.

**Requested Changes:**

Please address the weaknesses as critical changes, and the recommended changes are provided below.

R1. Empirical Scope:

Consider testing the prompt perturbations on a broader and more diverse set of domains beyond the current limitation of 5 benign technical topics to better support generalization.

R2. Human Evaluation:

Consider increasing the number of human annotators (currently limited to 2 for English and 4 for Japanese) and refining the annotation guidelines to establish a more reliable baseline for semantic perception.

R3. Z-Model Framework:

Consider simplifying the proposed 11-dimension Z-model to better align with the actual empirical experiments, as the current study only tests two highly localized pathways.

---

> ### Author Response · Authors · 2026-06-28
> **Response to Reviewer 4LQX**
>
> Dear Reviewer 4LQX,
>
> We sincerely thank you for the careful and balanced review. We are grateful that you recognized the controlled statistical design, the topic-aware hierarchical bootstrap and FDR handling, the transparency of the regex/prompt pipeline, and our restraint in separating mechanical marker breadth from reader-perceived sufficiency.
>
> We agree with your central concern: the evidence in this pilot supports protocol-level surface-marker redistribution, not reader-perceived semantic change, broad domain generalization, or empirical validation of the full 11-axis Z-model. In the revision, we therefore sharpened the claim boundary rather than expanding the paper into a calibrated semantic-perception study, which is outside the evidence-bearing claim of the present work.
>
> Regarding W1/R2, we agree that the low inter-reader agreement prevents the reader check from serving as confirmatory semantic validation. We revised Section 4.2 to make this explicit. The reader check is now framed as a boundary diagnostic, not as a validation instrument. The automatic regex and structural indicators are defined only as transparent, deterministic audit signals for surface-marker redistribution under matched perturbations. They are not treated as estimators of reader-perceived semantics, relevance-weighted sufficiency, pragmatic broadening, factuality, safety, or calibrated uncertainty. We also added future-work language noting that a larger direct-reader pool, structured pairwise comparisons, and annotator calibration would be required if the goal were to build a calibrated semantic-perception framework.
>
> Regarding W2/R3, we agree that the 11-axis presentation could appear over-engineered relative to the two audited pathways. We revised the Z-model presentation so that it is explicitly an observer-side reporting vocabulary rather than a validated latent-dimensional theory. Section 3 is now framed around observer-side reporting; stronger latent/interference terminology in the main empirical narrative has been replaced by operational secondary redistribution and observer-side reporting axes. We also added a Claim-Evidence Boundary table (Table 3) clarifying that the paper does not recover hidden states, prove a unique continuous latent space, or validate the full 11-axis reference basis. We retain the finer basis as a reporting resolution because reduced bases, especially 1D-2D, lose descriptive fit for the full proxy set (Section 3.2, Figure 2; Appendix L; Appendix K), while still disclaiming that 11 is minimal, unique, or geometrically meaningful. The intended contribution is accordingly narrower: a reporting vocabulary plus two localized audited pathways.
>
> Regarding W3, we agree that the Qwen result should be read as a model/snapshot-sensitivity signal rather than as cross-model validation. The open-weight path is now consistently described as a model-sensitivity comparison. The English HEDGE_EPI direction is clearer on the pinned open-weight checkpoint, whereas the Japanese COND increase is robust within the API-served pilot but essentially absent on Qwen/Qwen2.5-7B-Instruct. We therefore treat Japanese COND as an API-window marker-level finding, not as a model-family-invariant linguistic or pragmatic phenomenon.
>
> Regarding R1, we agree that five benign technical topics cannot support broad domain generalization. We therefore do not make such a claim. The topic set is described as a controlled pilot substrate, and broader domains, high-risk settings, adversarial prompts, dialogue tasks, and larger topic families are left to future work.
>
> Finally, we tightened the supplementary artifacts without changing any empirical conclusion: we added the pinned API snapshot identifier and run-level provenance metadata, and verified consistency between the manuscript and the supplementary bundle. A complete artifact-level changelog is provided in the "Changes Since Last Submission" note.
>
> In short, the revision is aligned with the evidence boundary identified in your review: the paper is a scoped diagnostic vocabulary and reproducible black-box protocol for auditing selected off-target marker redistributions, not a semantic validation study or a full theory of prompt-induced latent states.
>
> Sincerely,
> The Authors

---

### Author Response · Authors · 2026-06-28
**General Revision Note to the Action Editor and Reviewers**

Dear Action Editor and Reviewers,

We thank the reviewers for the constructive discussion and for helping us align the paper's claims with the evidence. In this revision, we made no new semantic-validation claim and did not add a new human-evaluation layer. Instead, following TMLR's evidence-matching criterion, we further narrowed the interpretation of the paper to a marker-level diagnostic pilot and strengthened the manuscript's claim boundaries, construct-validity discussion, and reproducibility artifacts.

The main changes are:

**1. Claim boundary and construct validity.** We further clarified that the automatic regex and structural proxies are used only as transparent indicators of surface-marker redistribution under matched perturbations. They are not treated as estimators of reader-perceived semantics, factuality, safety, or calibrated uncertainty. The direct-reader and LLM-judge checks are framed as boundary diagnostics, not as confirmatory semantic validation.

**2. Z-model presentation.** We softened the terminology around the Z-model and now present it as an observer-side reporting vocabulary rather than a validated 11-dimensional latent theory. The main text now foregrounds the four macro-groups and explicitly states that the paper does not recover hidden model states, prove a unique continuous latent space, or empirically validate the full 11-axis reference basis. We retain the finer-grained basis only as a reporting resolution justified by basis-reduction and rotation checks (Section 3.2, Figure 2; Appendix L), not as a minimality or dimensionality claim.

**3. Open-weight comparison.** The Qwen/Qwen2.5-7B-Instruct run is now consistently described as a model/snapshot-sensitivity comparison, not as cross-model validation. The Japanese COND effect is treated as a robust API-window marker-level finding, but not as a model-family-invariant linguistic or pragmatic phenomenon.

**4. Reproducibility and artifact consistency.** We tightened the supplementary bundle to strengthen the audit trail that underpins the marker-level claim: we added the pinned API snapshot identifier and run-level API provenance metadata, and verified consistency between the manuscript tables and the bundle. Manifest and checksum files were regenerated. These are artifact-hygiene improvements only; no reported conclusion changes. A complete artifact-level changelog is provided in the "Changes Since Last Submission" note.

We believe the revised manuscript now states the contribution at the correct level: a scoped diagnostic vocabulary plus a reproducible black-box protocol for auditing selected prompt-induced surface-marker redistributions, not a semantic-perception model or a full theory of prompt-induced latent states.

Sincerely,
The Authors